# SAMOSA: SHARPNESS AWARE MINIMIZATION FOR OPEN SET ACTIVE LEARNING

## ABSTRACT

Modern machine learning solutions require extensive data collection where labeling remains costly. To reduce this burden, open set active learning approaches aim to select informative samples from a large pool of unlabeled dataset that includes irrelevant or unknown classes. In this context, we propose Sharpness Aware Minimization for Open Set Active Learning (SAMOSA) as an effective querying algorithm. Building on theoretical findings concerning the impact that data typicality has on generalization properties of traditional SGD and sharpness-aware minimization, SAMOSA actively queries samples based on their typicality. SAMOSA effectively identifies atypical samples that belong to regions of the embedding manifold clustered close to the model decision boundaries. Therefore, SAMOSA prioritizes the samples which are both *(i)* highly informative for the targeted classes, and *(ii)* useful for distinguishing between targeted and unwanted classes. Extensive experiments show that SAMOSA achieves up to 3% accuracy improvement over the state-of-the-art across several datasets, while not introducing computational overhead. The source code of our experiments is available at `https://anonymous.4open.science/r/samosa-DAF4`.

## 1 INTRODUCTION

Modern machine learning models rely heavily on large labeled datasets, which are often costly or scarce (Liu et al., 2024; Weber et al., 2024; Schuhmann et al., 2022). In contrast, unlabeled data is more abundant but requires annotation (Zhao et al., 2022; Yang et al., 2024). Active Learning (AL) addresses this by selecting a small, informative subset of unlabeled data for labeling to improve model performance (Settles, 2009; Ren et al., 2021). However, in real-world scenarios, the unlabeled pool may include irrelevant samples from unknown classes (Yu et al., 2024). For instance, when training a mammal classifier, unlabeled data might include bird images, which can waste labeling resources as they are irrelevant to the target task. Practically, it has been observed that classic AL approaches perform poorly under such settings (Han et al., 2023).

Under such open-set scenario, there is an added layer of non-trivial finesse required for the data selection to be effective. Data selection in Open-Set Active Learning (OSAL) requires filtering out irrelevant samples while identifying informative, relevant ones from known classes Ning et al. (2022); Safaei et al. (2024); Park et al. (2022); Yang et al. (2023). Existing OSAL methods typically balance sample purity and informativeness by combining uncertainty-based criteria with mechanisms to distinguish known from unknown data points, often via clustering or specialized classifiers (Mao et al., 2024; Zong et al., 2024). While uncertainty-based sampling – often measured via metrics like entropy (Safaei et al., 2024) – focuses on selecting samples the model is least confident about, purity-based strategies prioritize samples that are highly representative of known classes, aiming to minimize the inclusion of out-of-distribution data (Ning et al., 2022). Both approaches have limitations: uncertainty-based methods overlook confidently misclassified atypical samples with low entropy, while purity-based methods tend to select typical, redundant examples that lie far from decision boundaries and add little new information. Moreover, recent findings by (Yang et al., 2023) highlight that informativeness – not strict in vs. out-of-distribution separation – is the primary driver of OSAL performance. Their work shows that sampling informative out-of-distribution (OOD) samples can significantly improve generalization and help refine decision boundaries in later rounds.

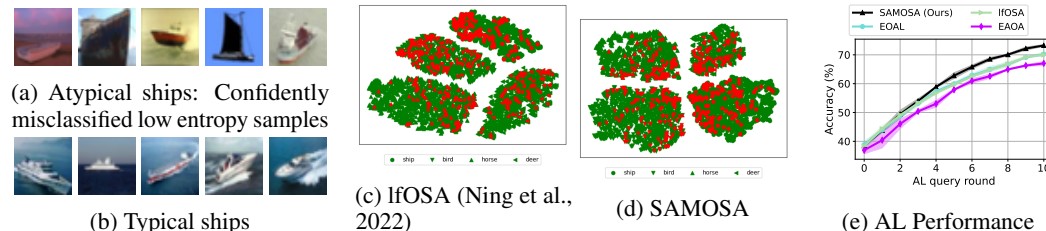

(a) Atypical ships: Confidently misclassified low entropy samples

(b) Typical ships

(c) lfOSA (Ning et al., 2022)

(d) SAMOSA

(e) AL Performance

Figure 1: Limitations of uncertainty-based sampling. **(a-b)** Visualization of: (a) lowest entropy samples that were misclassified for the class 'ship' on CIFAR10 with 30% mismatch at round 1. These are atypical, informative examples that will be ignored by the entropy criterion, but will be picked by our method; (b) batches of typical ships selected using entropy. **(c-d)** Embedding manifold visualization of the data points used to train (in green) against the queried sampled for labeling (in red) at round 5 over different approaches. Rather than relying on uncertainty or purity criterion which collect samples scattered across the classes clusters (c), our approach selects data points which fall close to decision boundaries, showcasing their informative nature (d). **(e)** Classification performance comparison on CIFAR100 with 40% mismatch ratio. SAMOSA overcomes state-of-the-art by 3%.

The importance of atypical samples has also been explored in interpretability literature (Kim et al., 2016), where representative examples are defined as prototypes and less typical examples as criticisms. In this context, *atypical criticisms* (e.g., white tigers) capture aspects of the data distribution that prototypes fail to represent, but still contribute meaningfully to the model's understanding and generalization. The inclusion of new data points alters the landscape of the model's understanding, potentially leading to revised representations of existing classes. This implies a dynamic adaptation of the model as it integrates new information, thereby refining the boundaries and characteristics of each class. Such reconfiguration of the model leads to new criticisms emerging from the remaining unlabeled data. Therefore, to leverage this dynamic adaptation and model reconfiguration in the OSAL domain, we propose to iteratively capture unlabeled data points that are atypical with respect to the current set of labelled examples, and that will in definite terms add to the model's generalization capabilities.

Many real-world datasets contain inherent subclasses – such as white tigers instead of typical yellow tigers, or objects in unusual contexts – that make certain data points atypical (e.g. see Feldman & Zhang (2020)[Fig. 1]). These atypical samples, though belonging to the same class, exhibit rare subpatterns and are often misclassified with high confidence Moon et al. (2020); Wei et al. (2022). As a result, in the OSAL context, entropy-based methods fail to capture them (see Figure 1a and Section 4), limiting generalization to rarer subpatterns. Similarly, purity-based strategies, while effective at filtering unknown classes, prioritize typical examples (Figure 1b) with strong purity scores, which do not lie near decision boundaries and thus contribute little to improving decision-making (see Figure 1c).

Building on top of the *atypical criticism* intuition and the findings on informativeness as the primary driver of OSAL performance (Yang et al., 2023), in this paper we propose Sharpness-Aware Minimization for Open-Set Active learning (SAMOSA). SAMOSA deliberately targets samples near decision boundaries (Figure 1d) – whether hard in-distribution cases or informative out-of-distribution instances – addressing the limitations of both uncertainty-based and purity-based approaches by prioritizing informativeness over superficial confidence or purity metrics. SAMOSA is grounded in the theoretical insight that the differential in predictive performance between stochastic gradient descent (SGD) and Sharpness-Aware Minimization (SAM) – which is a variant of the classic SGD designed to enhance model robustness – serves as a reliable indicator of atypicality. In contrast to existing OSAL approaches that explicitly aim to optimize sampling precision and recall, SAMOSA adopts a fundamentally different perspective by focusing on selecting highly atypical data points that entropy-based criteria often fail to detect (see Section 2). While this strategy may initially result in lower precision, it leads to a rapid increase in precision in subsequent rounds. Ultimately, SAMOSA achieves a recall that is comparable to or exceeds that of state-of-the-art methods, while consistently delivering superior model accuracy – as shown in Figure 1e – since atypical samples play a more

significant role in enhancing generalization (as shown in Section 3.2 and Garg & Roy (2023); Toneva et al. (2018)). A detailed experimental evaluation is available in Section 5.

**Contributions.** Our contributions are the following: *(i)* we consider relative atypicality as the key to effective OSAL subset selection; *(ii)* we design an effective and easily computable proxy for atypicality in OSAL; *(iii)* we present extensive empirical evaluation illustrating the importance of atypicality, the efficicacy of the proposed proxy metric, and achieve state-of-the-art performance for OSAL; and finally, *(iv)* we suggest a novel perspective on sample purity vs informativeness tradeoff in the field of OSAL.

## 2 MOTIVATION

In this section, we present a theoretical justification for using the differential between SGD and SAM. We consider a classification task with labels $y = \{\pm 1\}$ and input $\mathbf{x}$. Furthermore, we consider the setting where the data points belonging to each class $y$ contain several subclasses of $y$. This setting is inspired by natural datasets e.g. different breeds of a dog, or images of bottles sitting on a table vs being held by a human hand. Other subclasses can occur in the class distribution depending on spurious features, such as background information, orientation, and more.

We denote with $\mathbf{x}^{(y,z)}$ a sample data point belonging to the subclass $z$ of class $y$. We consider $z \in \{t, a\}$ denoting a subclass of typical and atypical samples in each class $y$. While in general there may be many typical and atypical subclasses within a class, we will focus on these two as they encapsulate the primary distinctions relevant to our study. The typical subclass $z = t$ captures the standard and more frequent characteristics expected within class $y$, whereas the atypical subclass $z = a$ highlights deviations from the typical subclass. As an illustrative example, consider the ship class in CIFAR10. There is abundant images of ferries compared to few images of kayaks. Images of ferries contain common features like the shape of the ferry, white color, or blue sea/ocean in the background while common features of kayaks compose of passengers with helmets, paddles, or creek in the background.

Say $L_D^{0-1}$ represent the 0-1 generalization error on the test data $D$. We use $\mathrm{D_{aty}}$ and $\mathrm{D_{typ}}$ to represent the subset of data with atypical and typical data points respectively. Let $\mathbf{W}_{\mathrm{SGD}}, \mathbf{W}_{\mathrm{SAM}}$ be the parameters representing the last 2 layers of a convolutional neural networks learnt using SGD and SAM respectively. We assume that the input to these 2 layers has *condensed* signals from various subclasses, so that the more frequent/typical subclass has a feature representation that contains more information about the class, which can be interpreted as a higher signal-to-noise ratio. The converse is true for the atypical subclass which has fewer data points, hindering the model from learning effective feature representations. We present the following proposition regarding the differential in their performances:

**Theorem 2.1.** *(Informal) Under the same regularity conditions and relative signal strengths of $t, a$, $\forall \epsilon > 0$, the difference in generalization losses for typical and atypical samples can be written as*

- $|L_{D_{typ}}^{0-1}(\mathbf{W}_{SAM}) - L_{D_{typ}}^{0-1}(\mathbf{W}_{SGD})| \leq \epsilon$

- $|L_{D_{aty}}^{0-1}(\mathbf{W}_{SAM}) - L_{D_{aty}}^{0-1}(\mathbf{W}_{SGD})| \geq 0.1 - \epsilon$

The proof and more formal setup with details are deferred to the Appendix A.6. Theorem 2.1 states that SGD and SAM exhibit similar generalization for typical data points, whereas their generalization performance differs for atypical data points. Consequently, the loss difference, or relevantly output probability difference, between SAM and SGD on unseen data points can distinguish typical and atypical samples.

We empirically verify this theorem in Section 3.2 by analyzing the position of samples in the embedding manifold of the model throughout model training and correlate it with the difference in SAM and SGD prediction scores for the same sample. In a nutshell, the experiments show that samples characterized by higher discrepancy between SAM and SGD predictions occupy the regions close to the decision boundaries in the embedding manifold, thus being more informative.

In the context of AL, it is relevant to query data points with lower signal strengths in the current training set. When labeled, these points boost the signal strength for their respective subclasses and

also bring in additional features/signals that are potentially relevant for another atypical subclass which had an even weaker signal strength before. This boost in the latter's signal may be enough for SAM to pick up, repeating the cycle to setup the stage for the next iteration of labeling. For example, whenever looking for images of bottles to be added to the labelled dataset, addition of images of bottles held by a human hand could provide a contextual signal boost for images showing bottles laying on a table with a human sitting next to it to be added in the next labeling. With this motivation and intuition in mind, we intend the differential between SGD and SAM as an effective iterative active learning sampler. Note that this discussion is relevant for expanding the model's understanding of both known as well as unknown labels in the unlabelled pool.

## 3 METHODOLOGY

In this section, we present our methodology for tackling the OSAL task. We first define the problem formulation in detail and give a background on the SAMIS-P metric. We then provide intuition on why relying on SAMIS scores for sampling data to be labelled in OSAL is effective. Finally, we present our SAMOSA algorithm.

**Problem Formulation:** Consider a labeled training dataset $\mathcal{D}_L$ with $K$ classes of interest (i.e. known classes) and a larger unlabeled dataset $\mathcal{D}_U$ with $K \cup U$ classes, where $U$ is the set of unknown irrelevant classes. The two sets are disjoint, i.e. $K \cap U = \emptyset$. Each labeled example $\mathbf{x}_i^L$ belongs to one of $K$ known classes, while an unlabeled example $\mathbf{x}_j^U$ could belong to either $K$ or $U$.

OSAL algorithms aim at iteratively selecting the set of samples to be labeled through a sequence of query rounds. In each round, the goal is to identify the most relevant samples $\mathcal{D}_Q \subset \mathcal{D}_U$ to be queried for annotation. While ideally we would want all chosen samples to only belong to $K$, often the selected queries are a mix of valid queries (i.e. the label of chosen data points belong to $K$) and invalid queries (i.e. the label of the data points belong to $U$). Formally, $\mathcal{D}_Q = X^K \cup X^U$, where $X^K = \{\mathbf{x}_i \mid label(\mathbf{x}_i) \in K, \mathbf{x}_i \in \mathcal{D}_Q\}$ and $X^U = \{\mathbf{x}_i \mid label(\mathbf{x}_i) \in U, \mathbf{x}_i \in \mathcal{D}_Q\}$. Total number of query budget per round is $q = |\mathcal{D}_Q|$. When the selected samples are annotated at the current round $t$, labeled and unlabeled datasets are updated as $\mathcal{D}_L^{t+1} = \mathcal{D}_L^t \cup X_t^K$ and $\mathcal{D}_U^{t+1} = \mathcal{D}_U^t \setminus \mathcal{D}_Q^t$. Because our method makes use of not only the valid queries $X^K$, but also invalid queries $X^U$, we also update a set of invalid queries $\mathcal{D}_{IQ}^{t+1} = \mathcal{D}_L^t \cup X_t^U, \mathcal{D}_{IQ}^0 = \emptyset$.

The goal of open-set active learning is to select informative queries within the constrained budget $q$ so that when target model $f_{test}$ is trained on the obtained labeled dataset, its performance in $|K|$-way classification task is maximized. Precision and recall metrics are commonly used to measure the quality of the OSAL sampling procedure, where precision is the fraction of valid queries over total queries and recall is the fraction of valid queries found so far over total number of known class samples. Formally, at each round $t$, $Precision_t = X_t^K / q_t$ and $Recall_t = |\mathcal{D}_L^t| / n_{known}$, where $n_{known}$ is the total number of samples belonging to the known classes.

### 3.1 SAMIS-P SCORES: MEASURING ATYPICALITY

Motivated by our intuitions in Section 2, we scrutinize the differential in SGD and SAM to identify atypical examples to be added to the labeled set in each iteration. We build on top of the findings made by (Yang et al., 2023) on sample informativeness being the key driver of performance in OSAL and conjecture that atypical samples are the most informative samples crucial for increasing target model performance incrementally.

Specifically, we define the SAMIS-P (SAM mInus Sgd-Probabilities) for our framework SAMOSA (described in the sequel) as the difference in output prediction class probabilities of the SAM and SGD models, making it an adequate approach for OSAL as it does not require the knowledge of the groundtruth label before query selection unlike previous works. Given an SGD trained model $f_{SGD}$ and a SAM trained model $f_{SAM}$ that output a probability vector, the SAMIS-P score function $S(\mathbf{x}_i)$ is defined as following:

$$S(\mathbf{x}_i) = \|f_{SAM}(\mathbf{x}_i) - f_{SGD}(\mathbf{x}_i)\|_1. \tag{1}$$

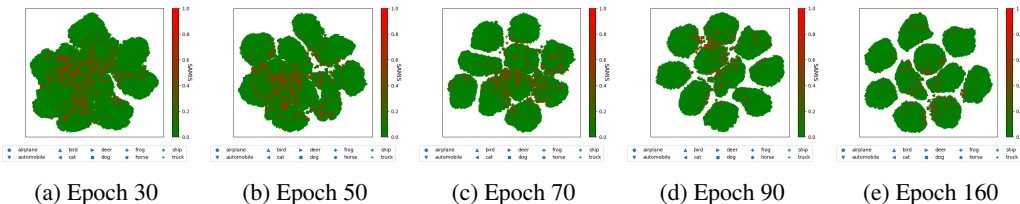

| (a) Epoch 30 | (b) Epoch 50 | (c) Epoch 70 | (d) Epoch 90 | (e) Epoch 160 |

Figure 2: Evolution of the impact of SAMIS-P scores of training samples on their embeddings across training epochs. Samples corresponding to high scores (red points) occupy critical positions relative to other data points, especially during the central epochs of the training where they are more concentrated around the boundary areas between the class clusters.

### 3.2 EFFICACY OF SAMIS-P

Here, we empirically verify our intuition of Section 2 illustrating the efficacy of the proposed SAMIS-P scores by showing the impact that the selected data samples have through training process of a neural network. More specifically, we study the position of samples in the embedding manifold of the model during training and correlate it with the samples' SAMIS-P scores. We train a ResNet18 model using SGD for 160 epochs on the CIFAR10 dataset and focus on the relationship between their SAMIS-P scores and their corresponding relative position in the embedding manifold.

Figure 2 shows the t-SNE visualization plot Van der Maaten & Hinton (2008) of the ResNet18 model embedding thorugh different epochs of the training procedure, along with the highest SAMIS-P scores data points. In the earlier part of the training ($< 50$ epochs), high SAMIS-P datapoints are placed in strategic areas of the embedding manifold, where multiple clusters are still joined together. The class boundaries are not learned yet. During the central portion of training – e.g. from epoch 50 up until epoch 120 – the highest SAMIS-P samples start moving towards occupying the border regions between the classes clusters in the embedding manifold. In the latter part when the training loss is close to 0 (beyond epoch 160), the high SAMIS-P score examples concentrate more towards learning data points that are too atypical and are basically singletons or outliers with little impact on generalization. These results support the intuition of Section 2 on the correlation between SAM vs SGD discrepancy and sample typicality. Therefore, we argue that using SAMIS-P scores for AL enables us to query informative data points effectively.

### 3.3 SAMOSA: SAMIS MEETS OSAL

Given the insights of Section 2 and findings of Section 3.2, we here propose Sharpness Aware Minimization for Open Set Active learning (SAMOSA) to effectively leverage SAMIS-P scores to identify the samples to be labeled in an OSAL setting. The SAMOSA algorithm (outlined in Appendix A.7) defines a two-step approach, in which known class samples are filtered from unknown class samples using a distinguisher NN in the first phase of the algorithm. In the second phase, atypical samples are selected from the known classes as the most informative samples to be labeled relying on SAMIS scores.

We train two models, $f_{SGD}$ and $f_{SAM}$, with cross-entropy loss using Stochastic Gradient Descent (SGD) and Sharpness-Aware Minimization (SAM). The models are initialized to have $|K|+1$ outputs, where the additional output dimension is for all unknown classes. After the first round, the queried samples compose of valid queries and invalid queries. In the future rounds, both the valid queries and invalid queries are added to be used as training set for the two models, where all the invalid queries receive a single label corresponding to the $|K| + 1_{th}$ dimension. At the querying step when training has finished, data samples from the unlabeled pool are passed through the models. Since the models learned to distinguish known from unknown class samples, the data points predicted as unknown class by SGD model *or* SAM model are rejected from further query selection.

In the process of carrying out rejection sampling, the output probabilities vector can be obtained for both models. We calculate the amount of disagreement between $f_{SGD}$ and $f_{SAM}$ following equation 1. Accepted samples from the unlabeled pool are sorted according to $S(x)$, and samples with the highest $S(x)$ – i.e., most atypical samples – are selected for querying and labelling. It is

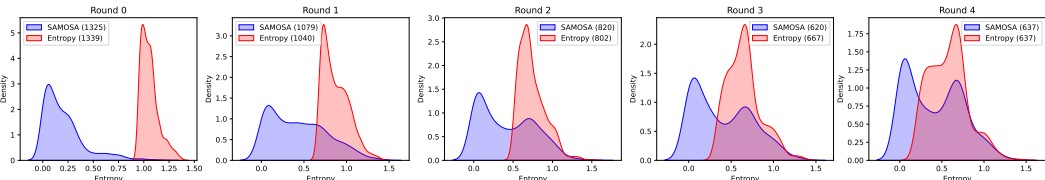

Figure 3: Entropy distribution of misclassified samples queried using high entropy (red) and high SAMIS (blue). Numbers in the legend indicate the number of misclassified samples. SAMOSA queries more samples the model misclassify with high confidence, especially in earlier rounds.

possible that on certain rounds, the number of accepted samples are less than total number of samples to query. In this case, remaining samples are chosen from the rejected samples with highest $S(x)$.

In the final step, the target model, $f_{test}$, is trained using cross-entropy loss and SGD on $\mathcal{D}_L$. In the real world scenario, this model classifies the known classes and is the model of interest. Because the unknown class samples are of irrelevance to the target model, it has $|K|$ outputs for only the known classes. At every round $t$, the target model is trained with labeled samples from the known classes obtained up to $t$ and is evaluated on the test set for performance.

**SAMOSA improves OOD filtering as AL rounds progress:** SAMOSA implicitly identifies the samples that contain useful information to distinguish between targeted and unwanted classes, selecting data points that belong to the decision boundaries between unwanted and target classes. These out-of-distribution samples will then be used to optimize the $K + 1$ filtering classifier in later OSAL rounds, providing highly relevant information on distinguishing between in-distribution (1 to $K$) and out-of-distribution ($K + 1$) samples. As a result, the $K + 1$ model will be more capable of filtering out unwanted classes as rounds progress. Practically, measuring disagreement between SAM and SGD predictions, SAMIS-P captures: *(i)* hard in-distribution samples that improve classification of known classes and boost final model performance, and *(ii)* informative out-of-distribution samples that strengthen the $K + 1$ classifier for future filtration, improving purity over time. We experimentally back this intuition in Appendix A.12.2, where we analyse the precision and recall of SAMOSA's sampling (lower precision/recall indicates higher number of out-of-distribution samples selected). While initially SAMOSA incurs in lower precision, it leads to a rapid increase in precision for subsequent rounds. Meanwhile, for later rounds, the sampling precision of SAMOSA increases, meaning that the querying process shift towards typical and clean data points—i.e., the $K + 1$ classifier has effectively learnt to distinguish between in- and out-of-distribution samples. Similarly, in Appendix A.8, we compare SAMOSA with its randomized counterpart which randomly selects unwanted class samples. Non-trivial drop in performance implies that atypical samples not only from the known classes, but also from unknown classes benefit the model substantially and is in accordance with Yang et al. (2023). Similarly, in Appendix A.9, we experimentally prove that the performance gains of SAMOSA are not simply due to an ensemble effect of leveraging multiple NNs for sampling, comparing SAMOSA against a disagreement-based sampling using three SGD models.

## 4 ENTROPY VS. TYPICALITY

Before delving into the direct comparison between SAMOSA and the state-of-the-art, we here verify limitations of entropy based criterion. More in detail, we verify that entropy based methods end up missing out on samples that the model misclassify with high confidence compared to SAMOSA. This represents a non-trivial limitation of entropy based methods because samples that the model misclassify confidently are crucial for improving model performance, more so than those that the model is slightly confused about. Such samples impact model training significantly in a positive sense, incurring high loss for the model to adjust its error when added to the training set. We run SAMOSA on CIFAR10 dataset with 40% of known classes. For each round, we select queries based on highest entropy values of the SGD model and highest SAMIS scores. From these selected queries, we only look at misclassified samples and their corresponding entropies.

The results in Figure 3 corroborate our concern as misclassified samples selected using high SAMIS scores contain low entropy samples, indicating the model is confident about its wrong predictions. On

| Strategy | CIFAR10 | | | CIFAR100 | | | TinyImageNet | | | Average Rank (↓) |
|---|---|---|---|---|---|---|---|---|---|---|
| | 20% | 30% | 40% | 20% | 30% | 40% | 20% | 30% | 40% | |
| Random | 96.25 ± 0.15 | 94.50 ± 0.15 | 92.33 ± 0.21 | 74.36 ± 0.38 | 68.40 ± 1.15 | 66.18 ± 0.06 | 39.34 ± 0.85 | 41.23 ± 0.72 | 38.68 ± 0.48 | 6.67 |
| BADGE | 97.26 ± 0.39 | 94.58 ± 0.62 | 92.76 ± 0.40 | 75.74 ± 0.46 | 66.00 ± 3.09 | 64.73 ± 2.15 | 38.73 ± 1.13 | 41.87 ± 0.89 | 38.80 ± 0.61 | 6.00 |
| LL | 95.76 ± 0.62 | 91.92 ± 0.86 | 86.16 ± 1.70 | 64.64 ± 1.18 | 58.37 ± 0.51 | 55.46 ± 1.38 | - | - | - | 11.33 |
| Coreset | 95.78 ± 0.30 | 92.22 ± 0.54 | 89.21 ± 0.70 | 70.00 ± 0.42 | 61.18 ± 1.01 | 57.88 ± 0.89 | 39.39 ± 1.73 | 40.62 ± 0.64 | 38.95 ± 0.89 | 8.22 |
| Entropy | 96.91 ± 0.20 | 94.03 ± 1.04 | 89.76 ± 0.97 | 69.89 ± 1.76 | 62.52 ± 1.25 | 62.13 ± 1.61 | 33.08 ± 0.99 | 36.53 ± 0.77 | 34.90 ± 0.40 | 8.56 |
| CONF | 96.24 ± 0.54 | 90.64 ± 1.61 | 85.35 ± 0.30 | 65.03 ± 0.86 | 57.93 ± 1.12 | 54.25 ± 0.51 | 35.64 ± 1.05 | 37.94 ± 0.53 | 35.38 ± 1.40 | 10.67 |
| Margin | 96.49 ± 0.67 | 92.62 ± 0.44 | 89.09 ± 0.87 | 68.36 ± 0.67 | 59.93 ± 1.11 | 57.20 ± 1.33 | 38.25 ± 0.61 | 40.70 ± 0.72 | 37.55 ± 0.65 | 8.89 |
| MQNet | 97.05 ± 0.22 | 93.82 ± 0.87 | 93.89 ± 0.28 | 75.94 ± 0.93 | 69.19 ± 0.52 | 67.38 ± 1.02 | - | - | - | 5.33 |
| lfOSA | 98.09 ± 0.22 | 96.17 ± 0.19 | 93.07 ± 0.65 | 82.94 ± 0.50 | 74.77 ± 0.42 | 70.00 ± 0.37 | 44.41 ± 1.10 | 45.35 ± 0.15 | 41.29 ± 0.43 | 3.33 |
| EOAL | 97.89 ± 0.27 | 96.49 ± 0.33 | 94.61 ± 0.25 | 81.78 ± 0.52 | 74.29 ± 0.37 | 70.25 ± 0.55 | 45.03 ± 0.84 | 45.76 ± 0.75 | 42.76 ± 0.71 | 2.89 |
| EAOA | 98.23 ± 0.07 | 96.56 ± 0.23 | 94.08 ± 0.18 | 79.75 ± 0.67 | 71.69 ± 0.26 | 67.03 ± 0.62 | 46.18 ± 0.11 | 46.75 ± 0.86 | 43.68 ± 1.39 | 2.67 |
| SAMOSA (Ours) | 98.00 ± 0.15 | 97.23 ± 0.18 | 96.19 ± 0.08 | 83.24 ± 0.47 | 75.75 ± 0.58 | 73.24 ± 0.30 | 46.38 ± 0.72 | 47.79 ± 0.34 | 43.22 ± 0.32 | 1.33 |

Table 1: Average accuracy and rank of $f_{test}$ after the last active learning rounds across each dataset and mismatch ratio setting. The best (second-best) approach for each setting is highlighted in green (blue). SAMOSA achieves the highest accuracy over most settings and has the best average rank.

the contrary, selection based on high entropy mainly selects samples the model misclassified with low confidence. This is more evident for earlier rounds – benefiting the model from early on – possibly due to the fact that the model trained with SAMOSA becomes less unsure about its predictions as rounds progress. We note that the total number of misclassified samples queried is similar across all rounds. Overall, these results prove the advantage of query selection using atypicality (SAMIS scores) over entropy (state-of-the-art).

## 5 EXPERIMENTS

We prove the effectiveness of SAMOSA, comparing our method against the state-of-the-art on three different datasets, namely CIFAR10, CIFAR100 (Krizhevsky et al., 2009), and TinyImageNet (Le & Yang, 2015). These datasets were chosen as they represent the golden standard in OSAL literature. We compare our method against 10 baselines that span standard AL and open-set AL literature. Namely, these include: *(i)* Entropy (Luo et al., 2013), *(ii)* Confidence (Lewis & Gale, 1994), *(iii)* Margin (Scheffer et al., 2001), *(iv)* CORESET (Sener & Savarese, 2018), *(v)* BADGE (Ash et al., 2020), *(vi)* LL (Yoo & Kweon, 2019), *(vii)* EOAL (Safaei et al., 2024), *(viii)* lfOSA (Ning et al., 2022), *(ix)* MQNet (Park et al., 2022), *(x)* EAOA (Zong & Huang, 2025), and *(xi)* random sampling (see Appendix A.2 for related work).

To enable a fair comparison, we follow the setup of Safaei et al. (2024) and Ning et al. (2022) and consider an AL procedure spanning through a total of 11 query rounds, in which 1500 samples from the unlabeled pool are queried according to each active learning algorithm at each round. For each dataset, mismatch ratios of 20%, 30%, and 40% are used, where mismatch ratio is the fraction of known classes $K$ over total number of classes $K + U$. Due to resource constraints, for the TinyImageNet dataset, we ignore comparison with the MQNet (Park et al., 2022) and LL (Yoo & Kweon, 2019) baselines since they do not represent tractably fast baselines and they did not complete the optimization process even after 10 GPU days. We ran our experiments using NVIDIA A100-80GB GPU over a span of more than 3000 GPU hours or equivalently 125 GPU days. More experimental details are made available in Appendix A.11. We also provide a detailed computational overhead analysis in Appendix A.10, showing that although SAMOSA requires training two models at each iteration, its computational requirements are comparable to the ones required by state-of-the-art approaches. For example, SAMOSA runs as quickly as EOAL and faster than MQNet.

**Results:** We present quantitative evaluation of our method against the baselines in Figure 4. We also present the average last round accuracy and its standard deviation for each OSAL method in Table 1. Additionally, the average rank of each method is reported, calculated as the average placement of the method against others for each experimental setting. The average rank metric serves as a quick tool to identify the best methodology as the one with the lowest rank (outperforming other baselines in most settings). SAMOSA achieves the highest accuracy over most settings and the lowest average rank, showcasing its superiority over the state-of-the-art. SAMOSA reaches up to $2\%$ accuracy improvements on CIFAR10 with 40% mismatch ratio, 3% on the same setting for CIFAR100 and 1% for the TinyImageNet dataset when the mismatch ratio is 30%.

SAMOSA's accuracy improvements over the baselines are directly proportional to the percentage of mismatch ratio. This behaviour is very desirable as the more challenging scenarios are indeed the ones with higher mismatch ratios. We argue that this is due to the ability of SAMOSA to select

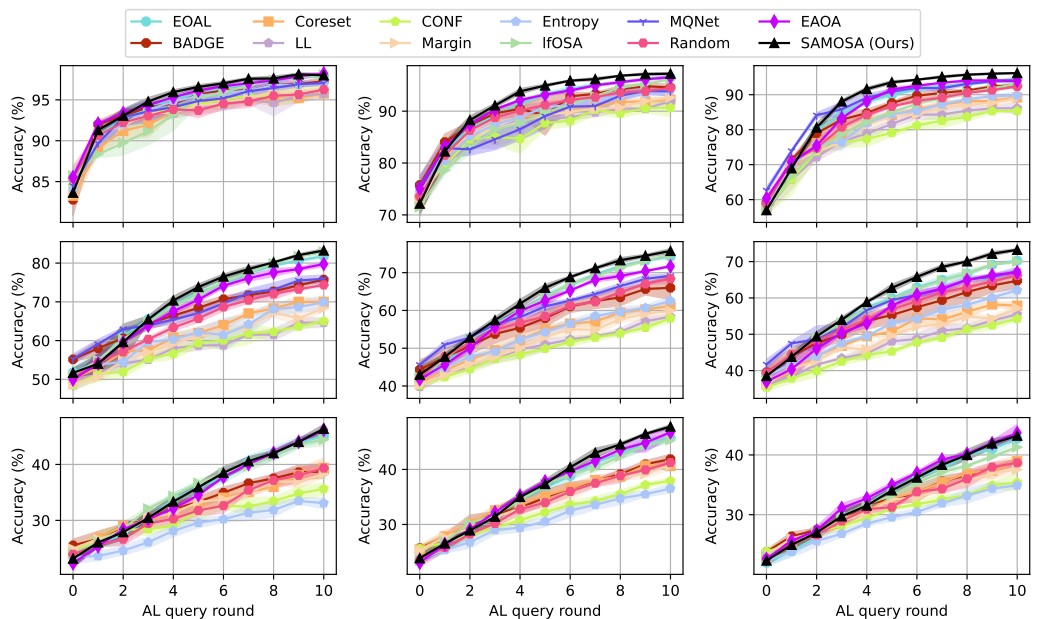

Figure 4: Classification performance comparison on CIFAR10 (first row), CIFAR100 (second row) and Tiny-Imagenet (third row) with 20% (first column), 30% (second column) and 40% (third column) mismatch ratio. SAMOSA overcomes state-of-the-art by up to 2% on CIFAR10, 3% on CIFAR100 and 2% on TinyImageNet.

samples which *(i)* are close to the decision boundaries of $f_{test}$, meaning that these samples are highly informative and that the more decision boundaries are present, the more the selected samples become informative; and *(ii)* are characterized by a high variety, meaning that each data point bring important unique information which is not redundant. We verify hypothesis *(i)* in Figures 12 to 14 of Appendix A.12.4, where we visualize how the data points selected at each round fall into the embedding space of the $f_{test}$ model for both SAMOSA and a purity-based OSAL approach like lfOSA. The data points collected using SAMOSA fall close to the $f_{test}$ model decision boundaries, showcasing their informative nature. By adding to the training set samples lying on the model decision boundaries, SAMOSA allows the AL process to better optimize the model decision-making, incurring in higher overall accuracy. On the other hand, samples collected using other approaches are scattered across the class clusters, bringing less information for each query round. Furthermore, we test hypothesis *(ii)* by visualizing the samples queried by SAMOSA for different rounds in Appendix A.12.3. The visualization confirms our intuition, selected images belong to atypical subclasses such as kayaks, cobles and even banana boats for the class ship of CIFAR10 (Figure 9b). Despite fewer labeled samples, SAMOSA performs better overall, highlighting that *sample quality is more crucial for model performance than the quantity of labeled samples*, supporting our original intuition.

**Label annotation noise:** Inspired by SAMOSA's improvement over higher mismatch ratios, we here consider to study its robustness against label noise during annotation. To this end, we consider a realistic scenario where the label annotator can make mistakes.

We consider a setting where for each round, 5% of the labels are randomly flipped before their addition to the dataset and compare SAMOSA against EOAL and LFOSA, two of the most popular and effective baselines. Table 2 reports the final test accuracies. Under label noise, SAMOSA largely outperforms the baselines, proving its robustness. This property is rooted in SAM's robustness to label noise Baek et al. (2024). Moreover, the baselines' performance drops drastically – especially LFOSA –, high-

Table 2: Final test accuracy of $f_{test}$ under label noise setup. SAMOSA largely outperforms the baselines.

| Method | CIFAR10 | | | CIFAR100 | | |
|--------|---------|---------|---------|----------|---------|---------|
| | 20% | 30% | 40% | 20% | 30% | 40% |
| SAMOSA | 95.10 | 95.00 | 93.38 | 77.90 | 70.90 | 69.90 |
| EOAL | 94.85 | 93.40 | 90.10 | 75.85 | 70.06 | 66.83 |
| LFOSA | 88.15 | 86.50 | 83.18 | 70.50 | 63.53 | 61.98 |

lighting how this real-world scenario has been previously
ignored by OSAL literature.

## 6  PRECISION VS. EFFECTIVENESS

Our experiments show that sample quality is more crucial for model performance (*effectiveness*) than the quantity of labeled samples (*precision*). Indeed, SAMOSA overcomes the state-of-the-art performance by focusing on selecting effective atypical samples rather than explicitly targeting precision/recall like other OSAL works do. Accordingly, we argue that precision and recall are indicative but not always desirable metrics and here analyse the *precision* vs. *effectiveness* trade-off.

We define a *bucketed* version of SAMOSA in which samples are selected from a bucket defined over a range of SAMIS scores rather than selecting those with the highest scores. Without loss of generality, we consider using 10 different – possibly overlapping – buckets. Bucket 1 contains samples with lowest SAMIS scores, while bucket 10 carries samples with highest SAMIS scores. Bucket indices from 2 to 9 contain samples with intermediate SAMIS values. The lower the bucket index, the more typical samples are queried, whereas higher bucket indices correspond to more atypical samples being selected. The bucketed SAMOSA defines an approach to examine a middle-ground between selecting the most typical or atypical data points.

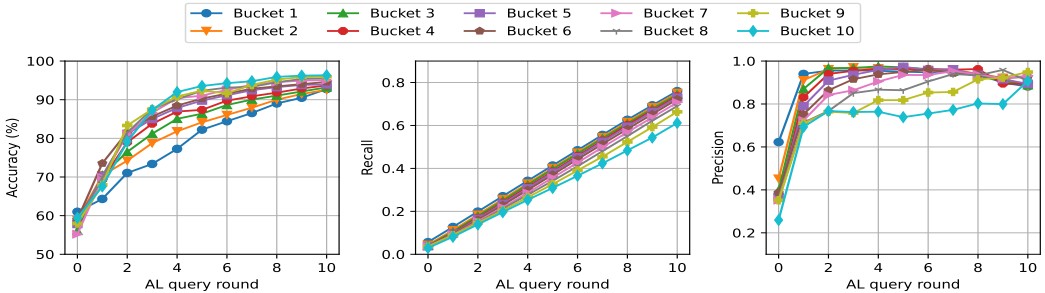

Figure 5: Accuracy, recall, and precision of bucketed SAMOSA over the CIFAR10 dataset with $40\%$ mismatch ratio. As bucket number increases, the precision drops while the accuracy increases.

We test the bucketed SAMOSA over the CIFAR10 dataset with $40\%$ mismatch ratio. Figure 5 shows the achieved accuracy, precision, and recall when selecting different bucket indices. The increasing order of bucket index translates directly to accuracy. That is, the higher the bucket index – equivalent to high SAMIS scores or atypical data points –, the higher the accuracy. On the other hand, the order of bucket number translates inversely to recall. Recall, which is calculated using total number of selected valid queries, drops as bucket number increases. Overall, these results show that there is no good trade-off between targeting the final *effectiveness* and targeting sampling *recall*. To achieve the highest accuracy, selecting only the most atypical samples is desirable. Meanwhile, constructing the largest dataset requires selecting only the most typical samples. None of the intermediate solutions can construct a larger dataset while achieving a higher accuracy.

**SAMOSA-L:** Driven by the findings in Figure 5, we introduce a variant of SAMOSA that excels in finding the most number of valid samples that belong to the known classes. While the primary goal of OSAL in current literature is model performance, we recognize that it may be of more interest for certain users to build a larger dataset of valid samples. In this case, recall would be the most important factor in analyzing return of investment for human annotation. To this end, we provide SAMOSA-L, which selects samples with the lowest $S(x)$ as opposed to the highest.

We compare SAMOSA-L against the state-of-the-art and the original version of SAMOSA in Appendix A.12.6. SAMOSA-L finds the highest number of known-class samples, especially for more challenging scenarios with higher mismatch ratios (see Figure 18). Meanwhile, it is interesting to observe that SAMOSA-L shows very competitive performance in accuracy especially for lower noise ratios (see Figure 16). We conjecture that this is dependent on the difficulty of the task. For tasks where typical samples suffice in classifying the known classes, SAMOSA-L performs well. For more challenging task with higher mismatch ratio (i.e. 40%), however, atypical samples are required

for distinguishing the classes effectively, leading to inferior performance. Overall, the empirical evaluations of SAMOSA and SAMOSA-L support the value that sample typicality brings to OSAL.

## 7 CONCLUSIONS

We propose SAMOSA the new state-of-the-art open-set AL algorithm based on a new proxy designed to capture atypicality on unlabelled datapoints. SAMOSA is theoretically grounded on the performance differential between SGD and SAM over atypical samples characterized by weaker signal. Through extensive experiments, we demonstrate the effectiveness of our algorithm compared with several exisiting baselines. In the future, we aim to find even faster way of computing SAMIS scores and characterizing sample atypicality without relying on signal assumptions.

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

# A APPENDIX

## A.1 LLM USAGE

LLMs were used to assist with grammar correction and sentence-level proofreading throughout the manuscript.

## A.2 RELATED WORK

**Active Learning** Active learning (AL) aims to select the most useful samples within a constrained query budget for label annotation. Standard AL assumes a closed-set scenario where all unlabeled data samples are in-distribution, belonging to one of the classes in the initial labeled dataset. Effective sampling strategies have been devised to select and label samples that would maximize model performance (Ash et al., 2020; Luo et al., 2013; Cho et al., 2024; Kothawade et al., 2021). These sampling strategies include using uncertainty measures such as entropy (Luo et al., 2013) and confidence (Lewis & Gale, 1994), using margin-based measures (Scheffer et al., 2001), training a loss prediction module (Yoo & Kweon, 2019), leveraging a deep Bayesian model (Gal et al., 2017), adopting a coreset framework (Sener & Savarese, 2018), contrastive learning (Du et al., 2023) and approximating the distance from the model decision boundaries (Cho et al., 2024). Disagreement-based sampling approaches Hanneke (2014); Fu et al. (2021); Seung et al. (1992) are also popular, where the outputs of a pool of models is compared to measure sample relevance. Lastly, SAAL Kim et al. (2023) leverages SAM to identify relevant samples to be labeled, using the samples pseudo-label to compute the max perturbation loss. SAMOSA extends the concept of disagreement-based approaches (Hanneke, 2014; 2007), integrating it with loss sharpness notions Kim et al. (2023), while targeting the more challenging open-set scenario and avoiding to rely on pseudo-labels.

**Open-Set Active Learning** Recently, researchers have geared attention towards Open-Set Active Learning (OSAL) where unlabeled samples can be out-of-distribution, belonging to unknown classes not present within the initial labeled set. Samples belonging to the unknown classes are considered useless to the target model. Some OSAL works focus on rejecting unknown-class samples by introducing a new model layer that estimates the probability of sample belonging to unknown classes (Bendale & Boult, 2016) or by using a Gaussian mixture model on the activation values (Ning et al., 2022). Similarly, (Zong et al., 2024) propose pushing the unknown class examples toward regions with high-confidence predictions and filter them using bidirectional uncertainty which jointly estimate uncertainty posed by both positive and negative learning. On the other hand, Park et al. Park et al. (2022) suggest a purity vs informativeness trade-off and propose a meta learning framework that balances selecting known-class samples and informative samples. Similarly, Authors in Safaei et al. (2024); Mao et al. (2024) propose combining entropy and inconsistency-based methods for informativeness and class-wise clustering. Interestingly, Yang et al. Yang et al. (2023) argue that selecting valuable unknown-class instances can improve model performance, hinting that informativity is more relevant than sampling precision. SAMOSA extends this intuition by providing identifying informativity through atypicality. Finally, the trade-off between purity and informativeness is also considered by (Zong & Huang, 2025), where the authors define an energy-based sampling framework that integrates epistemic uncertainty with aleatoric uncertainty.

## A.3 FURTHER DISCUSSION ON SAMOSA VS ENTROPY

In this section, we extend the discussion of comparing entropy and SAMIS based sampling and provide more detail on Figures 1a and 1b of the main text. SAMOSA was run as base algorithm for querying new samples. For every round, 4000 samples with lowest entropy were collected and of those 4000, we visually examined samples misclassified by the model.

We now discuss another set of experiments intended to examine advantage of our method over using entropy. After applying rejection sampling and only accepting samples classified as known class, samples are sorted according to their corresponding entropy. $3 * q$ number of samples with the highest entropy are selected for further selection, where $q$ is the total query budget per round. Within these high entropy samples, $q$ samples are selected for final query either from the lower end or higher end of SAMIS scores. We report the actual entropy values for some of selected queries at round 4 in Figure 6. The results display images with high entropy, but low SAMIS scores that are typical samples and images with lower entropy, but higher SAMIS that are atypical samples. These results support that SAMIS scores can provide information for querying valuable samples not obtainable by entropy. In later rounds, entropy does correlate better with SAMIS scores, becoming more indicative of sample typicality as the models become more reliable with larger labeled dataset.

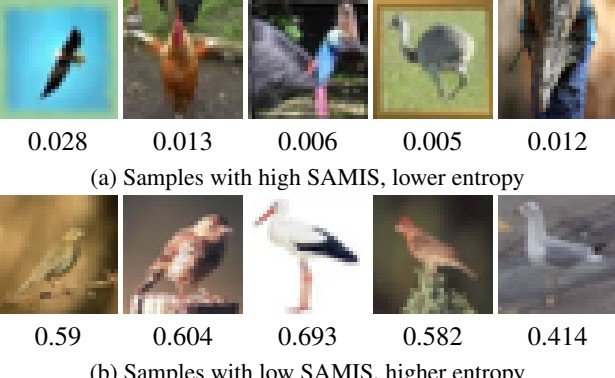

|  |  |  |  |  |
|---|---|---|---|---|
| 0.028 | 0.013 | 0.006 | 0.005 | 0.012 |

(a) Samples with high SAMIS, lower entropy

|  |  |  |  |  |
|---|---|---|---|---|
| 0.59 | 0.604 | 0.693 | 0.582 | 0.414 |

(b) Samples with low SAMIS, higher entropy

Figure 6: Visualization of data points with highest entropy queried using high SAMIS (top) and low SAMIS (bottom) scores and their corresponding entropy for the class 'bird' on CIFAR10 with 30% mismatch at round 4. Entropy is not always indicative of sample typicality, but is captured by SAMIS.

## A.4 Importance of Criticisms

Authors in Kim et al. (2016) proposed a novel method using maximum mean discrepancy (MMD) and witness function to find prototypes and criticisms. They define prototypes as representative samples that fit the model very well. Prototypes are not enough for obtaining best interpretability as real world data distributions are not 'clean' in the sense that it can be sufficiently represented by only representative samples. In this aspect, they suggest the importance of criticisms, which are samples that do not fit the model quite well, but occupy parts of the input space where prototypes do not provide good explanations. In a human subject pilot study, the value of criticisms are highlighted, achieving highest accuracy in user classification task when subjects were provided with both prototypes *and* criticisms.

The definition of prototypes and criticisms align quite well with typical and atypical samples discussed in this paper. Insights from their work provide an additional potential advantage of selecting atypical samples for annotation queries: Aiding human annotators for more accurate data annotation. Although perfect annotation accuracy was assumed for the purpose of this paper, methods for detecting and dealing with erroneous human annotation is an active area of research on its own. We leave the exploration of how our method can help human annotators perform better as future work.

## A.5 Sharpness Aware Minimization (SAM)

Here we introduce the concept and definition of Sharpness Aware Minimization (Foret et al., 2020). There have been a line of works following Foret et al. (2020) proposing an adaptive degree of sharpness (Kwon et al., 2021), using sparse perturbations for better stability and efficiency (Mi et al., 2022), and periodically applying gradient ascent step for further speed-up (Liu et al., 2022). For this paper, we consider the original setup of Foret et al. (2020).

SAM is an optimization algorithm designed to find a flatter minima. Consider a model $f : X \to Y$ parameterized by a weight vector $w$ and a per-sample loss function $l: W \times X \times Y \to R_+$. Given a sample S = $\{(x_1, y_1),..., (x_n, y_n)\}$ sampled i.i.d. from a data distribution D, traditional training loss is defined as $L_S(w) = \sum_{i=1}^{n} l(y_i, f(x_i, w))/n$. Sharpness Aware Minimization combines traditional loss with sharpness term to minimize the difference between maximum loss in the vicinity of the current minima. Formally, it is defined as the following for L2-norm:

$$L_{SAM}(w) = \min_w L_S(w) + [\max_{\|\epsilon(w)\|_2 \leq \rho} L_S(w + \epsilon(w)) - L_S(w)]$$

$$= \min_w \max_{\|\epsilon(w)\|_2 \leq \rho} L_S(w + \epsilon(w)),$$

$$\epsilon^*(w) \approx \rho \cdot sign\left(\nabla_w L_S(w)\right) \cdot \frac{|\nabla_w L_S(w)|}{\sqrt{\|\nabla_w L_S(w)\|_2}}$$

## A.6 Proof of Theorem 3.1

We borrow the notations and regulatory conditions as well as Lemmas A1-A4 from (Chen et al., 2023),

$P$ is number of patches for a CNN model, and $B$ is batch size.

*Condition* A.1. Suppose there exists a sufficiently large constant $C$, such that the following hold:

1. Dimension $d$ is sufficiently large: $d \geq \tilde{\Omega}\left( \max\{nP^{-2}\sigma_p^{-2}\|\boldsymbol{\mu}\|_2^2, n^2, P^{-2}\sigma_p^{-2}Bm\}\right)$.

2. Training sample size $n$ and neural network width satisfy $m, n \geq \tilde{\Omega}(1)$.

3. The 2-norm of the signal satisfies $\|\boldsymbol{\mu}\|_2 \geq \tilde{\Omega}(P\sigma_p)$.

4. The noise rate $p$ satisfies $p \leq 1/C$.

5. The standard deviation of Gaussian initialization $\sigma_0$ is appropriately chosen such that $\sigma_0 \leq \tilde{O}\left(\left(\max\left\{P\sigma_p d/\sqrt{n}, \|\boldsymbol{\mu}\|_2\right\}\right)^{-1}\right)$.

6. The learning rate $\eta$ satisfies $\eta \leq \tilde{O}\Big( \big( \max\big\{ P^2\sigma_p^2 d^{3/2}/(Bm), P^2\sigma_p^2 d/B, n\|\boldsymbol{\mu}\|_2/(\sigma_0 B\sqrt{d}m), $

$nP\sigma_p\|\boldsymbol{\mu}\|_2/(B^2 m\epsilon)\big\} \big)^{-1} \Big).$

**Lemma A.2.** *(Chen et al., 2023)[Informal statement of lemmas A.3 and A.4] Let $p$ be the strength of the label flipping noise. For any $\epsilon > 0$, under certain regularity conditions, with high probability, there exists $0 \leq t \leq T$ such that the training loss converges, i.e., $L_S(\mathbf{W}^{(t)}) \leq \epsilon$. Besides,*

1. ***For SGD***, *when the signal strength $\|\boldsymbol{\mu}\|_2 \geq \Omega(d^{1/4})$, we have $L_{\mathcal{D}}^{0-1}(\mathbf{W}^{(t)}) \leq p + \epsilon$. When the signal strength $\|\boldsymbol{\mu}\|_2 \leq O(d^{1/4})$, we have $L_{\mathcal{D}}^{0-1}(\mathbf{W}^{(t)}) \geq p + 0.1$.*

2. ***For SAM***, *provided the signal strength $\|\boldsymbol{\mu}\|_2 \geq \tilde{\Omega}(1)$, we have $L_{\mathcal{D}}^{0-1}(\mathbf{W}^{(t)}) \leq p + \epsilon$.*

**Lemma A.3.** *(Chen et al., 2023) For any $\epsilon > 0$, under Condition A.1, with probability at least $1 - \delta$ there exists $t = \tilde{O}(\eta^{-1}\epsilon^{-1}mnd^{-1}P^{-2}\sigma_p^{-2})$ such that:*

1. *The training loss converges, i.e., $L_S(\mathbf{W}^{(t)}) \leq \epsilon$.*

2. *When $n\|\boldsymbol{\mu}\|_2^4 \geq C_1 dP^4\sigma_p^4$, the test error $L_{\mathcal{D}}^{0-1}(\mathbf{W}^{(t)}) \leq p + \epsilon$.*

3. *When $n\|\boldsymbol{\mu}\|_2^4 \leq C_3 dP^4\sigma_p^4$, the test error $L_{\mathcal{D}}^{0-1}(\mathbf{W}^{(t)}) \geq p + 0.1$.*

**Lemma A.4.** *(Chen et al., 2023) For any $\epsilon > 0$, under Condition A.1 with $\sigma_0 = \tilde{\Theta}(P^{-1}\sigma_p^{-1}d^{-1/2})$, choose $\tau = \Theta\Big( \frac{m\sqrt{B}}{P\sigma_p\sqrt{d}} \Big)$. With probability at least $1 - \delta$, neural networks first trained with SAM with $O\Big( \eta^{-1}\epsilon^{-1}n^{-1}mB\|\boldsymbol{\mu}\|_2^{-2} \Big)$ iterations, then trained with SGD with $\tilde{O}\Big( \eta^{-1}\epsilon^{-1}mnd^{-1}P^{-2}\sigma_p^{-2} \Big)$ iterations can find $\mathbf{W}^{(t)}$ such that,*

1. *The training loss satisfies $L_S(\mathbf{W}^{(t)}) \leq \epsilon$.*

2. *The test error $L_{\mathcal{D}}^{0-1}(\mathbf{W}^{(t)}) \leq p + \epsilon$.*

Now, define a classification task with label $y \in \{\pm 1\}$. We further define each class as a composition of two subclasses, with $\alpha\boldsymbol{\mu}$ and $\beta\boldsymbol{\mu}$ being signal strengths for typical and atypical subclass, respectively.

**Definition A.5** (Our Data Setup). Let $\boldsymbol{\mu} \in \mathbb{R}^d$ be a fixed vector representing the signal contained in each data point. Each data point $(\mathbf{x}, y)$ with input $\mathbf{x} = [\mathbf{x}^{(1)\top}, \mathbf{x}^{(2)\top}, \ldots, \mathbf{x}^{(P)\top}]^\top \in \mathbb{R}^{P\times d}, \mathbf{x}^{(1)}, \mathbf{x}^{(2)}, \ldots, \mathbf{x}^{(P)} \in \mathbb{R}^d$, the true label $y \in \{-1, 1\}$ is generated from one of two distributions $\mathcal{D}_{typ}$ and $\mathcal{D}_{aty}$ specified as follows:

1. A noise vector $\boldsymbol{\xi}$ is generated from the Gaussian distribution $\mathcal{N}(\mathbf{0}, \sigma_p^2\mathbf{I})$, where $\sigma_p^2$ is the variance.
2. For $\mathcal{D}_{typ}$, one of $\mathbf{x}^{(1)}, \mathbf{x}^{(2)}, \ldots, \mathbf{x}^{(P)}$ is randomly selected and then assigned as $y \cdot \alpha \cdot \boldsymbol{\mu}$, which represents the signal, while the others are given by $\boldsymbol{\xi}$, which represents noises.
3. For $\mathcal{D}_{aty}$, one of $\mathbf{x}^{(1)}, \mathbf{x}^{(2)}, \ldots, \mathbf{x}^{(P)}$ is randomly selected and then assigned as $y \cdot \beta \cdot \boldsymbol{\mu}$, which represents the signal, while the others are given by $\boldsymbol{\xi}$, which represents noises.

and training set $S$ is drawn from $\mathcal{D}_{typ}$.

**Theorem A.6** (formal statement of Theorem 3.1). *Under the conditions of Lemma A.3 and Lemma A.4, models trained using SGD and SAM+SGD achieve training loss $L_S(\mathbf{W}_{SGD}^{(t)}) \leq \epsilon$ and $L_S(\mathbf{W}_{SAM}^{(t)}) \leq \epsilon$ at the respective times specified in the lemmas. There exists $\alpha$ and $\beta$ such that*

1. *SGD generalizes well on the samples from one subclass of $y$ but not the other, meaning $L_{D_{typ}}^{0-1}(\mathbf{W}_{SGD}^{(t)}) \leq \epsilon$ and $L_{D_{aty}}^{0-1}(\mathbf{W}_{SGD}^{(t)}) \geq 0.1$.*

2. *SAM generalizes well on both subclasses of $y$, meaning $L_{D_{typ}}^{0-1}(\mathbf{W}_{SAM}^{(t)}) \leq \epsilon$ and $L_{D_{aty}}^{0-1}(\mathbf{W}_{SAM}^{(t)}) \leq \epsilon \quad \forall i, y$.*

3. *$|L_{D_{typ}}^{0-1}(\mathbf{W}_{SAM}^{(t)}) - L_{D_{typ}}^{0-1}(\mathbf{W}_{SGD}^{(t)})| \leq \epsilon$ and $|L_{D_{aty}}^{0-1}(\mathbf{W}_{SAM}^{(t)}) - L_{D_{aty}}^{0-1}(\mathbf{W}_{SGD}^{(t)})| \geq 0.1 - \epsilon \quad \forall i, y$.*

Choose

$$\alpha \geq \frac{P\sigma_P}{\|\mu\|_2} \sqrt[4]{\frac{C_1 d}{n}} \geq \frac{\tilde{\Omega}(1)}{\|\boldsymbol{\mu}\|},$$

$$\frac{P\sigma_P}{\|\mu\|_2} \sqrt[4]{\frac{C_3 d}{n}} \geq \beta \geq \frac{\tilde{\Omega}(1)}{\|\boldsymbol{\mu}\|}$$

We first prove item 1 of Theorem A.6.

*Proof.*

$$\alpha \geq \frac{P\sigma_P}{\|\mu\|_2} \sqrt[4]{\frac{C_1 d}{n}} \quad \rightarrow \quad n\|\alpha\boldsymbol{\mu}\|_2^4 \geq C_1 d P^4 \sigma_p^4$$

Then by lemma A.3,

$$L_{D_{typ}}^{0-1}(\mathbf{W}_{SGD}^{(t)}) \leq \epsilon \tag{2}$$

$$\beta \leq \frac{P\sigma_P}{\|\mu\|_2} \sqrt[4]{\frac{C_3 d}{n}} \quad \rightarrow \quad n\|\beta\boldsymbol{\mu}\|_2^4 \leq C_3 d P^4 \sigma_p^4$$

Then by lemma A.3,

$$L_{D_{aty}}^{0-1}(\mathbf{W}_{SGD}^{(t)}) \geq 0.1 \tag{3}$$

$\square$

Now we prove item 2.

*Proof.* Since $\beta \geq \frac{\tilde{\Omega}(1)}{\|\boldsymbol{\mu}\|}$, $\|\beta\boldsymbol{\mu}\| \geq \tilde{\Omega}(1)$ and since $\alpha \geq \frac{\tilde{\Omega}(1)}{\|\boldsymbol{\mu}\|}$, $\|\alpha\boldsymbol{\mu}\| \geq \tilde{\Omega}(1)$. Then by lemma A.4,

$$L_{D_{typ}}^{0-1}(\mathbf{W}_{SAM}^{(t)}) \leq \epsilon \quad \text{and} \quad L_{D_{aty}}^{0-1}(\mathbf{W}_{SAM}^{(t)}) \leq \epsilon \tag{4}$$

$\square$

Now we prove item 3.

*Proof.* Since $L_{D_{typ}}^{0-1}(\mathbf{W}_{SAM}^{(t)}) \leq \epsilon$ and $L_{D_{typ}}^{0-1}(\mathbf{W}_{SGD}^{(t)}) \leq \epsilon$, then $|L_{D_{typ}}^{0-1}(\mathbf{W}_{SAM}^{(t)}) - L_{D_{typ}}^{0-1}(\mathbf{W}_{SGD}^{(t)})| \leq \epsilon$.

Meanwhile, since $L_{D_{aty}}^{0-1}(\mathbf{W}_{SGD}^{(t)}) \geq 0.1$ and $L_{D_{aty}}^{0-1}(\mathbf{W}_{SAM}^{(t)}) \leq \epsilon$, then $|L_{D_{aty}}^{0-1}(\mathbf{W}_{SAM}^{(t)}) - L_{D_{aty}}^{0-1}(\mathbf{W}_{SGD}^{(t)})| \geq 0.1 - \epsilon$. $\square$

## A.7 SAMOSA ALGORITHM

---

**Algorithm 1** SAMOSA Algorithm

---

1: **Input:**
   Labeled data $\mathcal{D}_L$, unlabeled data $\mathcal{D}_U$, number of query rounds $R$, known classes $K$, queries-per-round $q$, models $f_{SGD}, f_{SAM}, f_{test}$
2: **Process:**
3: $\mathcal{D}_{IQ} \leftarrow \emptyset$                    # *Initial invalid queries*
4: **for** $r = 0, 1, \ldots, R - 1$ **do**
5:     $\mathcal{D}_Q \leftarrow \emptyset$               # *selected queries for this round*
6:     Train $f_{SAM}$ and $f_{SGD}$ on $\mathcal{D}_L \cup \mathcal{D}_{IQ}$
7:     Train $f_{test}$ on $\mathcal{D}_L$ and evaluate performance
8:     $\forall x \in \mathcal{D}_U$, compute $S(x)$ following Equation (1)
9:     $\mathcal{D}_{\text{reject}} \leftarrow \{x \in \mathcal{D}_U \mid x \text{ is predicted as unknown by } f_{SAM} \text{ or } f_{SGD}\}$
10:    Sort $\mathcal{D}_U \setminus \mathcal{D}_{reject}$ in descending $S(x)$
11:    $\mathcal{D}_Q \leftarrow$ the first q elements from the sorted set
12:    Obtain $X^k$ and $X^u$ from $\mathcal{D}_Q$        # *Valid/invalid queries*
13:    $\mathcal{D}_L \leftarrow \mathcal{D}_L \cup X^k, \mathcal{D}_{IQ} \leftarrow \mathcal{D}_{IQ} \cup X^u$
14:    $\mathcal{D}_U \leftarrow \mathcal{D}_U \setminus \mathcal{D}_Q$            #*Update datasets*
15: **end for**

---

## A.8 SAMIS IMPROVES DETECTION OF UNWANTED CLASSES

In this section, we analyze more in detail the effectiveness of the sampling process relying on SAMIS scores for improving the filtering capabilities of the $K + 1$ distinguisher network which is in charge of classifying samples coming from unwanted classes. To this end, we design SAMOSA-R as an alternative version of SAMOSA where, for each round, the same number of queried irrelevant class samples were randomly selected as opposed to selection according to high SAMIS-P. We compare the SAMOSA-R performance against the SAMOSA approach for the most challenging setups of 40% noise level on the CIFAR10 and CIFAR100 datasets in Table 2.

SAMOSA outperforms SAMOSA-R, corroborating its ability to query valuable irrelevant class samples. Intuitively, out-of-distribution samples selected for labelling will be used to optimize the $K + 1$ filtering classifier in later OSAL rounds. Samples selected using SAMOSA will belong to regions of the embedding manifold clustered close to the decision boundaries, bringing highly relevant information on distinguishing between in-distribution (1 to $K$ classes) and out-of-distribution ($K + 1$ class) samples. As a

Table 2: Performance comparison of SAMOSA against its randomized version (SAMOSA-R). SAMOSA queries valuable irrelevant class samples, improving the filtering capabilities of the $K + 1$ distinguisher network.

| Strategy | CIFAR10 (40%) | CIFAR100 (40%) |
|----------|---------------|----------------|
| SAMOSA | $96.19 \pm 0.08$ | $73.24 \pm 0.3$ |
| SAMOSA-R | $95.58 \pm 0.29$ | $70.91 \pm 0.51$ |

result, for later rounds, the $K + 1$ model will be more capable of filtering out OOD samples. Figure 7 backs this intuition showing how SAMOSA achieves high sampling precision relying only on high informativeness, meaning that highly informative samples improve the performance of the $K + 1$ filtering model as well. Thus, SAMOSA provides an effective way of improving both the performance of the distinguisher model and the final model by selecting highly informative samples that lie near decision boundaries. Our results are in line with Yang et al. (2023), highlighting that careful selection of not only relevant samples, but irrelevant (OOD) samples matter as well.

## A.9 SAMOSA IS NOT ONLY AN ENSEMBLE EFFECT

In this section, we analyze more in detail the root cause of the performance boost achieved by SAMOSA, aiming at investigating if the performance benefits derive solely from the ensemble effect of using two NNs for the sampling process. To this end, we compare SAMOSA against a disagreement-based method adapted to the OSAL setting in

Table 4: Comparison of running times between SAMOSA and the state-of-the-art. SAMOSA is as fast, if not faster, than most state-of-the-art approaches.

| CIFAR100 | 20% noise (H:M:S) | 30% noise (H:M:S) | 40% noise (H:M:S) |
|---|---|---|---|
| SAMOSA (ours) | 7:21:34 | 8:48:51 | 10:20:32 |
| EOAL | 7:47:48 | 9:07:14 | 10:49:02 |
| LFOSA | 4:14:22 | 5:10:52 | 6:08:21 |
| MQNet | 15:11:29 | 15:34:52 | 16:17:03 |

which three SGD models are ensembled and samples are queried based on prediction disagreement, following Algorithm 0 in Hanneke (2014) combined with $K + 1$ filtration. Table 3 shows the results of the comparison between the disagreement-based performance against the SAMOSA approach for the most challenging setups of 40% noise level on the CIFAR10 and CIFAR100 datasets. The disagreement-based approach combined with $K + 1$ filtration does prove to be an effective strategy, but still underperforms compared to SAMOSA. This result highlights the additional benefits that SAMOSA brings to the disagreement-based approach and proves that the effectiveness of SAMOSA is rooted on the performance difference between SAM and SGD on atypical samples as proven in Theorem 3.1.

Table 3: Performance comparison of SAMOSA against disagreement-based sampling. SAMOSA's benefits are rooted on the SAM and SGD differential of Theorem 3.1.

| Strategy | CIFAR10 (40%) | CIFAR100 (40%) |
|---|---|---|
| SAMOSA | $96.19 \pm 0.08$ | $73.24 \pm 0.3$ |
| Disagreement | $95.63 \pm 0.21$ | $72.39 \pm 0.19$ |

### A.10 SAMOSA's Computational Overhead

In this section, we analyze in details the computational overhead required to run SAMOSA and compare it against state-of-the-art approaches. we quantitatively measure the approach running time and present the time required for top performing SOTA methods including SAMOSA, EOAL, LFOSA, and MQNet on CIFAR100. For fair evaluation, we used an NVIDIA H100 GPU across all methods. The obtained results are shown in Table 4.

The results indicate that our method does not incur higher computational overhead compared to state-of-the-art approaches. SAMOSA runs as quickly as EOAL and converges to the final solution much quicker than the MQNet baselines, which are highly regarded approaches in the OSAL domain. The only approach quicker than SAMOSA is LFOSA, which achieves 3% less accuracy. Therefore, we believe that SAMOSA represents a very competitive approach when compared to the available counterparts. Therefore, while it is true that SAMOSA relies on training two models, its computational requirements are comparable to the ones required by state-of-the-art approaches. Indeed, the baselines contain components that increase the computational overhead over simple SGD training as well. For example, EOAL involves clustering algorithm, LFOSA involves training of Gaussian Mixture Models, and MQNet involves self supervised learning combined with meta-learning.

We also provide a theoretical analysis of SAMOSA's time complexity followingly. Define $Q_c^{(t)}$ and $Q_u^{(t)}$ the samples queried at round $t$ for the relevant and irrelevant classes respectively. $O_{SGD}$ and $O_{SAM}$ the time required to run a single optimization step for SGD and SAM. $L$ is the time required to label a single sample. $S^{(0)}$ is the initial training set size. For query round $t$, the complexity of running SAMOSA is approximable to:

$$O_{SGD}(\sum_i^{t-1} Q_c^{(i)} + Q_u^{(i)} + S^{(0)}) + O_{SAM}(\sum_i^{t-1} Q_c^{(i)} + Q_u^{(i)} + S^{(0)}) + Q_c^t L + O_{SGD}(\sum_i^{t-1} Q_c^{(i)} + S^{(0)}).$$

$$(5)$$

Considering that $O_{SAM} \simeq 1.5 \cdot O_{SGD}$, $S^{(0)} < \sum_i^T Q_c^{(i)} < sum_i^T Q_c^{(i)} + Q_u^{(i)}$, $Q_c^{(i)} + Q_u^{(i)} = Q$ by definition, and $Q_c^{(i)} < Q$, the total time complexity of running SAMOSA for T rounds is bounded by the following equation:

$$SAMOSA \simeq \mathcal{O}(3.5T^2 Q O_{SGD} + TQL). \tag{6}$$

Similarly, considering EOAL and MQNet, we can define $C$ as the time complexity of the clustering step of EOAL and $\alpha$ as a complexity multiplier for the optimization step of the second MQNet model. The EOAL and MQNet complexity can then be derived as:

$$EOAL \simeq \mathcal{O}(T^2 Q(2O_{SGD} + C) + TQL), \tag{7}$$

$$MQNet \simeq \mathcal{O}(T^2 Q(2 + \alpha)O_{SGD} + TQL). \tag{8}$$

In practice, $\alpha > 1.5$ and $C \simeq 1.5 \cdot O_{SGD}$ hold, thus showing that SAMOSA runs faster than MQNet and comparably to EOAL (as backed by our experiments).

### A.11 ADDITIONAL EXPERIMENTAL DETAILS

The anonymized source code used to implement SAMOSA and run all experiments is made publicly available and can be found at `https://anonymous.4open.science/r/samosa-EF8C`. For instructions on how to run the code, interested readers should reference the README file.

We run our experiments on the three well-known CIFAR10, CIFAR100 (Krizhevsky et al., 2009), and TinyImageNet datasets (Le & Yang, 2015). CIFAR10 and CIFAR100 consist of $32 \times 32$ images from 10 and 100 classes, respectively. These datasets both contain 50 thousand training samples and 10 thousand test images. TinyImageNet is a larger dataset consisting of $64 \times 64$ images coming from 200 classes and contains 100 thousand training samples and 10 thousand test images.

In our setting, we consider an AL procedure spanning through a total of 11 query rounds. At query round 0, all methods start with an initial labeled training set composed of a percentage of samples randomly selected from known classes. To enable a fair comparison, we follow the setup of Safaei et al. (2024) and Ning et al. (2022) and set this random initial percentage to 1% for CIFAR10 and 8% for CIFAR100 and TinyImageNet. Every round, 1500 samples from the unlabeled pool are queried according to each active learning algorithm. The test set for active learning composes of all the samples that belong to the known classes in the original datasets' test set.

For fair evaluation, same model architecture and training mechanism are enforced on $f_{test}$ across all methods. We use the model architecture used by Park et al. (2022) for $f_{test}$. For models used for selecting queries, ResNet18 (He et al., 2016) is used as the common base model architecture. Modifications to the model architecture are specific to each algorithm, and we adopt their code for the models. For each dataset, mismatch ratios of 20%, 30%, and 40% are used, where mismatch ratio is the fraction of known classes $K$ over total number of classes $K + U$. For instance, 30% mismatch ratio indicates 3 known classes for CIFAR10, 30 for CIFAR100, and 60 for TinyImageNet. For reliability of the results, we report the average and standard deviation over 4 runs using different random seeds. The four random seeds used were randomly set to 1, 15, 42, and 85.

Here we provide training details of our models and $f_{test}$ across all baselines. We use ResNet18 architecture with dropout 0.2 and train each model for 300 epochs using batch size 128. For SGD and SAM, we set momentum 0.9, weight decay 5e-4, and initial learning rate 0.01. We employ step learning rate scheduler with step size 60 and gamma 0.5.

For SAM optimizer, we consider the most widely used setting and apply L2-norm for $\epsilon(w)$ and set $\rho = 0.05$ Foret et al. (2020). Meanwhile, for the MQNet baseline, we trained CSI model for 100 epochs.

### A.12 ADDITIONAL EXPERIMENTAL RESULTS

### A.12.1 DETAILED NUMERICAL COMPARISON

In this section, we provide a detailed numerical comparison between the performance achieved by SAMOSA and SAMOSA-L against state-of-the-art. Table 1, Tables 5 and 6 present the accuracy achieved by each OSAL method on all experimental settings considered in the paper after the last, eighth and fifth sample selection round.

| Strategy | CIFAR10 | | | CIFAR10 | | | TinyIamgeNet | | | Average Rank (↓) |
|---|---|---|---|---|---|---|---|---|---|---|
| | 20% | 30% | 40% | 20% | 30% | 40% | 20% | 30% | 40% | |
| Random | $95.49 \pm 0.22$ | $93.62 \pm 0.23$ | $90.55 \pm 0.57$ | $72.05 \pm 1.41$ | $64.85 \pm 0.31$ | $62.94 \pm 0.33$ | $37.16 \pm 1.26$ | $38.84 \pm 1.22$ | $35.99 \pm 1.38$ | 6.67 |
| BADGE | $96.80 \pm 0.27$ | $94.25 \pm 0.64$ | $91.18 \pm 0.23$ | $72.70 \pm 0.79$ | $63.47 \pm 2.22$ | $61.64 \pm 2.69$ | $37.51 \pm 0.77$ | $39.12 \pm 0.63$ | $36.62 \pm 0.53$ | 5.67 |
| LL | $94.64 \pm 1.58$ | $90.50 \pm 0.44$ | $85.46 \pm 1.97$ | $61.53 \pm 1.47$ | $54.37 \pm 0.87$ | $51.66 \pm 1.15$ | - | - | - | 11.33 |
| Coreset | $94.88 \pm 0.52$ | $91.77 \pm 0.20$ | $88.16 \pm 0.38$ | $68.31 \pm 0.82$ | $59.43 \pm 0.76$ | $56.16 \pm 1.52$ | $35.96 \pm 0.32$ | $38.69 \pm 0.70$ | $36.92 \pm 0.32$ | 8.33 |
| Entropy | $96.79 \pm 0.35$ | $92.87 \pm 0.84$ | $88.68 \pm 0.39$ | $68.14 \pm 1.71$ | $59.67 \pm 1.31$ | $58.11 \pm 0.30$ | $31.88 \pm 1.22$ | $34.73 \pm 1.10$ | $33.19 \pm 1.26$ | 8.56 |
| CONF | $95.48 \pm 0.95$ | $89.60 \pm 0.81$ | $83.73 \pm 1.30$ | $62.30 \pm 0.66$ | $53.94 \pm 0.63$ | $51.40 \pm 0.99$ | $33.45 \pm 0.73$ | $35.52 \pm 0.72$ | $33.29 \pm 0.38$ | 10.67 |
| Margin | $95.96 \pm 0.48$ | $90.70 \pm 1.36$ | $87.12 \pm 0.37$ | $67.20 \pm 1.23$ | $57.74 \pm 0.90$ | $54.87 \pm 1.32$ | $36.91 \pm 0.47$ | $38.57 \pm 0.52$ | $35.55 \pm 0.83$ | 9.00 |
| MQNet | $96.45 \pm 0.32$ | $92.99 \pm 0.89$ | $93.10 \pm 0.41$ | $73.15 \pm 1.75$ | $66.43 \pm 0.70$ | $65.11 \pm 0.73$ | - | - | - | 5.17 |
| IfOSA | $97.45 \pm 0.24$ | $94.52 \pm 0.47$ | $90.58 \pm 0.51$ | $79.94 \pm 0.68$ | $71.35 \pm 0.52$ | $66.79 \pm 0.94$ | $41.51 \pm 0.93$ | $42.37 \pm 0.53$ | $38.19 \pm 0.15$ | 3.44 |
| EOAL | $97.28 \pm 0.26$ | $95.52 \pm 0.06$ | $93.03 \pm 0.13$ | $79.35 \pm 0.23$ | $71.81 \pm 0.20$ | $66.70 \pm 0.59$ | $41.96 \pm 1.10$ | $42.88 \pm 0.88$ | $40.03 \pm 0.34$ | 3.00 |
| EAOA | $97.38 \pm 0.28$ | $95.60 \pm 0.21$ | $94.05 \pm 0.35$ | $77.56 \pm 0.98$ | $69.12 \pm 0.72$ | $64.93 \pm 0.17$ | $42.05 \pm 0.59$ | $43.61 \pm 0.33$ | $40.19 \pm 0.13$ | 2.67 |
| SAMOSA (Ours) | $97.61 \pm 0.29$ | $96.83 \pm 0.07$ | $95.73 \pm 0.19$ | $80.20 \pm 0.06$ | $73.28 \pm 0.68$ | $70.06 \pm 0.31$ | $42.01 \pm 0.44$ | $44.55 \pm 0.32$ | $39.99 \pm 0.89$ | 1.33 |

Table 5: Average accuracy of $f_{test}$ after *eight* active learning rounds across each dataset and mismatch ratio setting. Similarly to Table 1 we report the average accuracy, its standard deviation and the average rank of each method. The best (second-best) approach for each setting is highlighted in green (blue). SAMOSA achieves the highest accuracy over most settings and has the best average rank.

| Strategy | CIFAR10 | | | CIFAR10 | | | TinyIamgeNet | | | Average Rank (↓) |
|---|---|---|---|---|---|---|---|---|---|---|
| | 20% | 30% | 40% | 20% | 30% | 40% | 20% | 30% | 40% | |
| Random | $93.71 \pm 0.87$ | $91.19 \pm 0.59$ | $86.67 \pm 0.29$ | $65.71 \pm 1.01$ | $58.38 \pm 1.71$ | $56.89 \pm 0.46$ | $31.78 \pm 0.95$ | $33.95 \pm 0.56$ | $31.26 \pm 0.57$ | 7.22 |
| BADGE | $95.66 \pm 0.56$ | $89.18 \pm 3.80$ | $87.63 \pm 0.45$ | $68.45 \pm 1.20$ | $57.96 \pm 2.28$ | $55.29 \pm 2.24$ | $33.33 \pm 1.01$ | $34.90 \pm 1.32$ | $32.58 \pm 0.89$ | 5.89 |
| LL | $94.03 \pm 0.72$ | $88.53 \pm 0.51$ | $81.64 \pm 1.01$ | $58.74 \pm 1.21$ | $50.99 \pm 1.57$ | $48.06 \pm 0.72$ | - | - | - | 11.17 |
| Coreset | $94.26 \pm 0.65$ | $89.43 \pm 0.46$ | $84.26 \pm 0.43$ | $61.94 \pm 1.55$ | $55.00 \pm 0.90$ | $50.79 \pm 1.11$ | $33.01 \pm 0.97$ | $35.18 \pm 1.22$ | $32.79 \pm 1.14$ | 7.67 |
| Entropy | $95.50 \pm 0.22$ | $90.77 \pm 0.29$ | $84.48 \pm 0.97$ | $62.13 \pm 0.94$ | $54.27 \pm 0.94$ | $53.08 \pm 0.38$ | $29.58 \pm 0.88$ | $30.56 \pm 0.96$ | $29.58 \pm 0.65$ | 8.11 |
| CONF | $94.46 \pm 0.34$ | $87.70 \pm 1.17$ | $79.13 \pm 1.01$ | $59.34 \pm 0.96$ | $49.88 \pm 0.44$ | $45.29 \pm 0.73$ | $31.81 \pm 1.82$ | $32.22 \pm 0.38$ | $31.02 \pm 0.23$ | 10.33 |
| Margin | $94.31 \pm 0.80$ | $89.25 \pm 0.48$ | $84.03 \pm 1.35$ | $61.26 \pm 1.17$ | $53.00 \pm 0.90$ | $49.64 \pm 0.58$ | $33.31 \pm 0.77$ | $33.79 \pm 1.05$ | $32.57 \pm 0.54$ | 8.67 |
| MQNet | $94.86 \pm 0.65$ | $89.03 \pm 2.33$ | $90.86 \pm 0.81$ | $67.08 \pm 0.85$ | $61.04 \pm 1.20$ | $59.41 \pm 0.45$ | - | - | - | 5.83 |
| IfOSA | $95.21 \pm 0.74$ | $89.74 \pm 0.81$ | $85.04 \pm 0.36$ | $71.96 \pm 0.83$ | $63.47 \pm 0.69$ | $59.69 \pm 0.40$ | $36.70 \pm 0.58$ | $37.32 \pm 0.32$ | $33.24 \pm 0.78$ | 4.00 |
| EOAL | $95.63 \pm 0.36$ | $93.27 \pm 0.38$ | $89.96 \pm 0.64$ | $72.50 \pm 1.40$ | $63.87 \pm 0.73$ | $59.95 \pm 0.54$ | $35.99 \pm 0.74$ | $36.67 \pm 0.52$ | $34.24 \pm 0.83$ | 2.67 |
| EAOA | $96.14 \pm 0.42$ | $93.18 \pm 0.21$ | $91.44 \pm 0.74$ | $70.55 \pm 0.53$ | $62.62 \pm 1.33$ | $57.88 \pm 0.26$ | $34.53 \pm 0.25$ | $37.82 \pm 0.27$ | $34.99 \pm 0.44$ | 2.89 |
| SAMOSA (Ours) | $96.59 \pm 0.24$ | $94.93 \pm 0.25$ | $93.56 \pm 0.41$ | $73.83 \pm 0.77$ | $65.99 \pm 0.87$ | $62.77 \pm 0.98$ | $35.91 \pm 0.76$ | $37.35 \pm 0.83$ | $34.06 \pm 0.71$ | 1.56 |

Table 6: Average accuracy of $f_{test}$ after *five* active learning rounds across each dataset and mismatch ratio setting. Similarly to Table 1 we report the average accuracy, its standard deviation and the average rank of each method. The best (second-best) approach for each setting is highlighted in green (blue). SAMOSA achieves the highest accuracy over most settings and has the best average rank.

Overall, SAMOSA achieves the highest accuracy over most experimental settings and the lowest average rank, showcasing its superiority over the state-of-the-art. SAMOSA reaches up to 2% accuracy improvements on CIFAR10, 3% on CIFAR100 and 2% on TinyImageNet. Moreover, SAMOSA is stably the best approach whenever considering more than 5 AL rounds. Therefore, SAMOSA represents the new state-of-the-art for open-set active learning.

Lastly, we note that SAMOSA's accuracy improvements over the baselines are directly proportional to the percentage of mismatch ratio and argue that this is due to the ability of SAMOSA to select samples close to the decision boundaries of $f_{test}$. In Appendix A.12.4, we delve more in detail into this hypothesis, proving its truthfulness. Briefly, the data points queried with SAMOSA fall close to the $f_{test}$ model decision boundaries especially during the earlier AL rounds, showcasing their informative nature. Therefore, SAMOSA allows for better optimisation of the model decision-making, incurring higher overall accuracy.

### A.12.2 PRECISION AND RECALL

Here, we report the precision and recall for our method and the baselines. Figure 7 presents the results for sampling precision, while Figure 8 shows the sampling recall results.

Overall, SAMOSA-L consistently achieves the highest sampling recall across all datasets and settings, proving its effectiveness in identifying clean, typical data points belonging to the targeted known classes. More specifically, SAMOSA-L largely outperforms the baselines for sampling precisions during the first AL iterations. Meanwhile, for later rounds, the SAMOSA-L's precision slightly decreases being outperformed by few other approaches on some settings. This trend is tightly connected with the accuracy trend analysed in Appendix A.12.1, with the reason behind this phenomenon once again being rooted in SAMOSA-L focusing on the selection of clean samples to ensure their belonging to known classes. Sampling clean typical data points (Appendix A.12.3) which do not lie close to the model decision boundaries (Appendix A.12.4), SAMOSA-L quickly identifies a large number of known classes samples from the very first AL rounds. Meanwhile, later on, the sampling precision of SAMOSA-L decreases, meaning that it already selected most of the highly

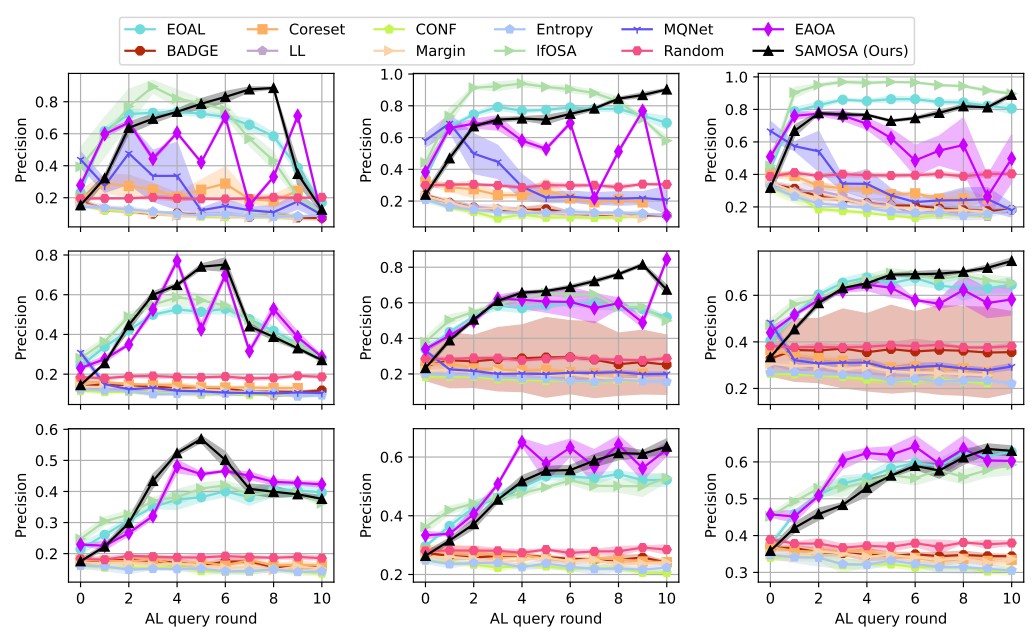

Figure 7: Sampling precision comparison on CIFAR10 (first row), CIFAR100 (second row) and Tiny-Imagenet (third row) with 20% (first column), 30% (second column) and 40% (third column) mismatch ratio.

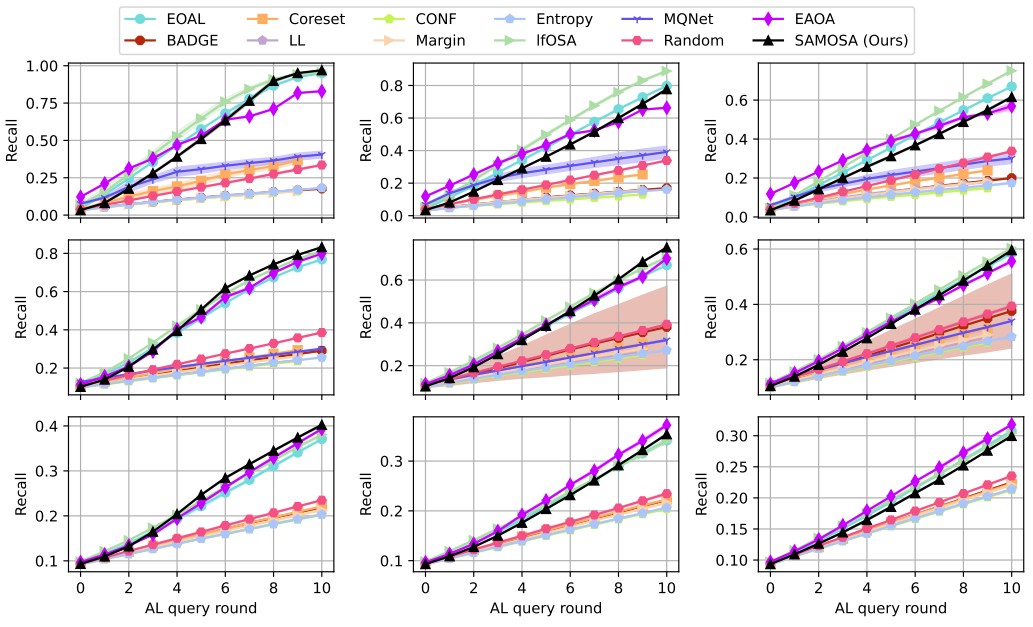

Figure 8: Sampling recall comparison on CIFAR10 (first row), CIFAR100 (second row) and Tiny-Imagenet (third row) with 20% (first column), 30% (second column) and 40% (third column) mismatch ratio.

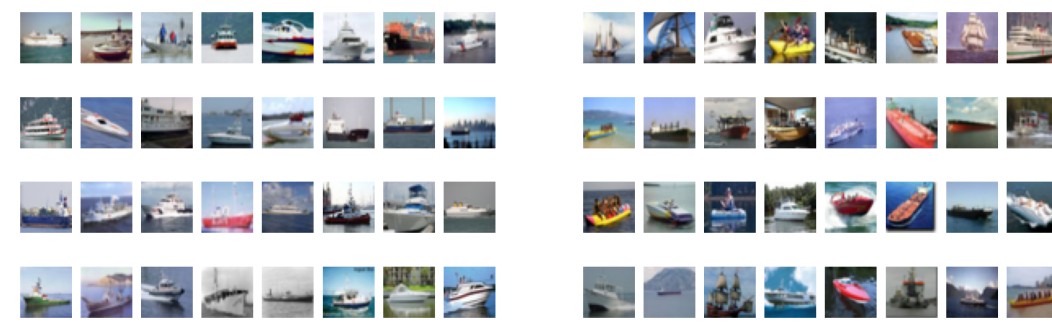

(a) Samples queried by SAMOSA-L at round 1          (b) Samples queried by SAMOSA at round 1

Figure 9: Comparison of the samples queried by SAMOSA-L and SAMOSA belonging to the class *ship* of the CIFAR10 datasets after the *first* AL round. Images selected by SAMOSA-L are more typical and clean samples of ships, in which a ship or boat is in the foreground, water is clearly visible, as well as the sky. Meanwhile, the images selected by SAMOSA are more atypical, containing kayaks, sailing ships, brigs, cobles and even banana boats. This proves our hypothesis on the SAMOSA-L behaviour during first AL iterations.

typical data points, thus making the querying process shift towards atypical samples, as verified in the embedding manifold analysis of Appendix A.12.4.

Lastly, it is relevant to notice that while SAMOSA outperforms several baselines in precision and recall, it falls short against lfOSA and EOAL in several settings. This trend is especially valid for the first AL rounds. We hypothesize that this behaviour is caused by SAMOSA focusing on querying atypical, highly informative samples, which sometimes may not belong to the targeted known classes. Meanwhile, for later rounds, the sampling precision of SAMOSA increases, meaning that it already selected most of the highly atypical samples, thus making the querying process implicitly shift towards typical and clean data points. Once again, we test this hypothesis in Appendices A.12.3 and A.12.4, proving its validity. Despite fewer labelled samples, SAMOSA performs better overall, once again highlighting that sample quality is more crucial for model performance than the number of labelled samples.

### A.12.3   QUERIED SAMPLE VISUALIZATION

Here, we visualize the data points queried by both SAMOSA and SAMOSA-L for each AL round. To recall, in Sections 6 and 7, we hypothesize that SAMOSA-L focuses on selecting clean samples to ensure their belonging to known classes, meaning that given the skewed distribution of data points (unbalanced towards clean and easy to classify samples), during the first few rounds SAMOSA-L selects similar data points that bring redundant information. Only whenever the number of rounds increases the variety of the selected samples is sufficient to produce meaningful model updates. On the other hand, we hypothesize that SAMOSA samples atypical data points – bringing relevant information for model updates –, thus incurring in high variety of samples also during the first few AL rounds. To test our hypothesis, we visualize the data points queried by both SAMOSA and SAMOSA-L for each AL round. Without loss of generality, we focus on the CIFAR10 datasets and set the mismatch ratio to 40%. Figures 9 to 11 show the queried samples over the class *ship* for the rounds number one, five and nine. Similar results can be achieved for classes different from ship or different datasets and mismatch ratio and are omitted to avoid redundancy.

After the *first* round, the images selected by SAMOSA-L are more typical and clean samples of ships, in which a ship or boat is in the foreground, water is clearly visible, as well as the sky, constructing the classic image of a ship in the middle of the sea (Figure 9a). Meanwhile, the images selected by SAMOSA are more atypical, containing kayaks, sailing ships, brigs, cobles and even banana boats (Figure 9b). These findings provide a partial proof of our hypothesis on the SAMOSA-L behaviour during the beginning of the OSAL process.

As the OSAL process continues some atypical samples starts to emerge in the data points selected by SAMOSA-L, such as the banana boat image (Figure 10a). However, the largest portion of queried

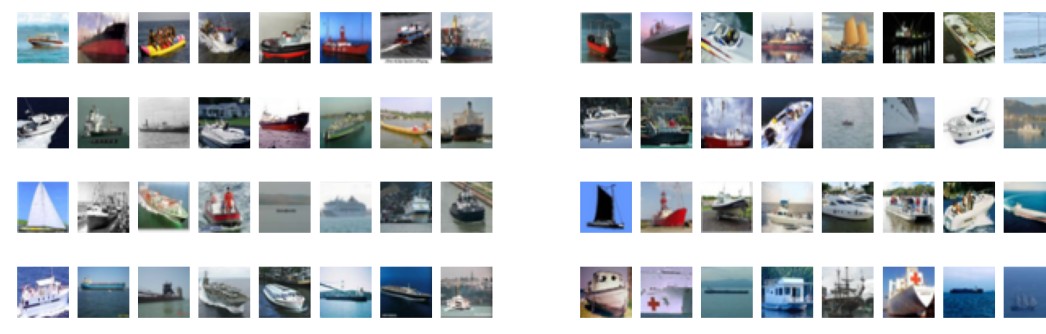

(a) Samples queried by SAMOSA-L at round 5          (b) Samples queried by SAMOSA at round 5

Figure 10: Comparison of the samples queried by SAMOSA-L and SAMOSA belonging to the class *ship* of the CIFAR10 datasets after the *fifth* AL round. After the first AL iterations some atypical samples starts to emerge in the data points selected by SAMOSA-L. However, the largest portion of queried samples are still highly typical, showing clean images of ships in the foreground with still water and/or blue sky. On the other hand, data points selected through SAMOSA are still largely atypical, being effective for training the NN model.

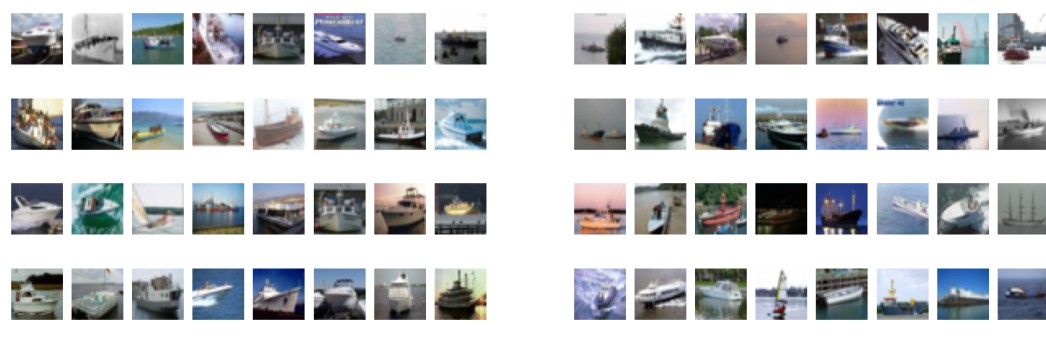

(a) Samples queried by SAMOSA-L at round 9          (b) Samples queried by SAMOSA at round 9

Figure 11: Comparison of the samples queried by SAMOSA-L and SAMOSA belonging to the class *ship* of the CIFAR10 datasets after the *ninth* AL round. During the last AL iterations, a good variety of typical and atypical samples starts to emerge in the data points selected by SAMOSA-L. This is due to the skewed distribution of data points that commonly occur in natural datasets. This proves our hypothesis on the SAMOSA-L behaviour during the later rounds.

samples are still highly typical, showing clean images of ships in the foreground with still water and/or blue sky. On the other hand, data points selected through SAMOSA are still largely atypical (Figure 13c), being effective for training the NN model.

Finally, during the latest AL iterations, a good variety of typical and atypical samples starts to emerge in the data points selected by SAMOSA-L (Figure 11a), caused by the skewed data distribution. These findings provide a partial proof of our hypothesis on the SAMOSA-L behaviour during the later rounds of the OSAL process. Moreover, these insights are aligned with the clear performance boost that SAMOSA-L achieves during the latest AL rounds as discussed in Section 6 and Appendix A.12.1.

Overall, the sample visualization confirms our original hypothesis on the SAMOSA-L behaviour and provides a solid backing of our insights in Section 6.

### A.12.4    SAMOSA EMBEDDING SPACE ANALYSIS

In Section 7, we argue that precise sampling may not always lead to effective model training, since not all training data points share the same level of relevance for the model optimization procedure. Relying on the value that sample typicality brings to open-set AL, we hypothesize that SAMOSA identifies atypical informative samples to be queried for labeling, which are helpful for enhancing

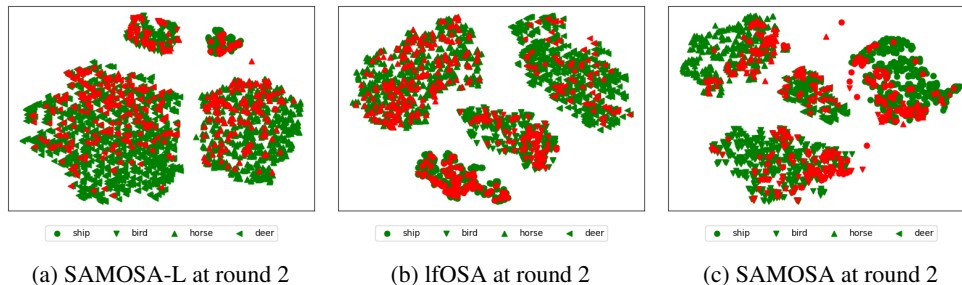

(a) SAMOSA-L at round 2          (b) lfOSA at round 2          (c) SAMOSA at round 2

Figure 12: Embedding manifold visualization of the data points used to train $f_{test}$ against the queried sampled for labeling at round 2 over different sampling approaches. Green points represent the training data, while red points represent the queried images to be used to optimize $f_{test}$ at the next iteration. The data points collected using SAMOSA fall close to decision boundaries, showcasing their informative nature. Meanwhile, samples collected using other approaches are scattered across the classes clusters.

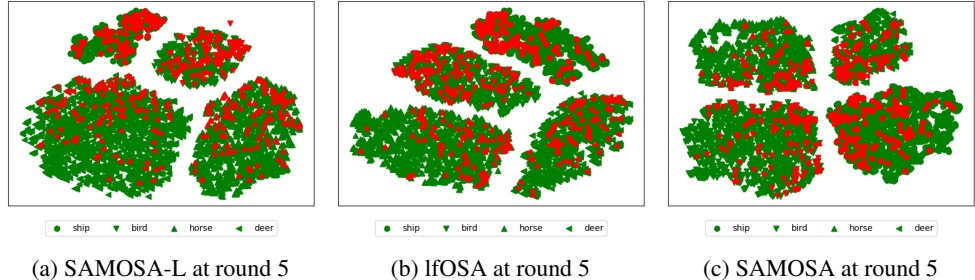

(a) SAMOSA-L at round 5          (b) lfOSA at round 5          (c) SAMOSA at round 5

Figure 13: Embedding manifold visualization of the data points used to train $f_{test}$ against the queried sampled for labeling at round 5 over different sampling approaches. Green points represent the training data, while red points represent the queried images to be used to optimize $f_{test}$ at the next iteration. The data points collected using SAMOSA fall close to decision boundaries, showcasing their informative nature. Meanwhile, samples collected using other approaches are scattered across the classes clusters.

the final model performance. Thus, in defining SAMOSA, we hypothesize that atypical samples are highly informative and represents the best choice for improving the effectiveness of the active learning process. In this section, we test this hypothesis by looking at the embedding manifold of $f_{test}$ over several AL rounds and for few sampling approaches. More in detail, we plot the embedding manifold for the $f_{test}$ model trained at different sampling steps (similar to Figure 2) and show where the embeddings of the data points sampled in the next iteration of an OSAL approach fall. Figures 12 to 14 show the results of our experiments for rounds two, five and eight. Similar results are achieved for all rounds and are omitted to avoid redundancy.

During the first AL rounds, it is clear that the data points queried and labeled using SAMOSA fall close to the model decision boundaries. The proximity of queried samples to the decision boundaries highlights their informative nature, as their use in a later optimization step allows the $f_{test}$ model to define a more accurate classification pattern. By adding to the training set those data points that lie close to the model decision boundaries, SAMOSA allows the AL optimization process to better take into account corner cases and inputs which are difficult to classify. Moreover, the placement of SAMOSA's selected data points in the embedding space explains the rapid accuracy improvements achieved by SAMOSA compared to the baselines for the earlier rounds (see Figure 4). On the other hand, samples collected using other approaches (SAMOSA-L and lfOSA in Figures 12 and 13) are scattered across the classes clusters. These samples may still be useful to increase the overall information available in the training dataset. However, they are less relevant for defining how to correctly separate the dataset classes, given their higher distance from the decision boundaries of the model.

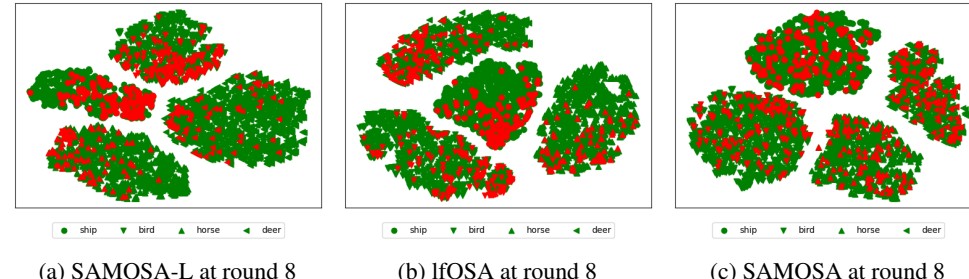

| (a) SAMOSA-L at round 8 | (b) lfOSA at round 8 | (c) SAMOSA at round 8 |

Figure 14: Embedding manifold visualization of the data points used to train $f_{test}$ against the queried sampled for labeling at round 8 over different sampling approaches. Green points represent the training data, while red points represent the queried images to be used to optimize $f_{test}$ at the next iteration. For later rounds, the difference between SAMOSA and other approaches is less evident. Some of the data points collected using SAMOSA-L and lfOSA fall close to decision boundaries. However, some of these decision boundaries may not be the most informative ones, as it happens for the bottom cluster in (a) and for the top cluster in (b).

Finally, it is interesting to notice that for later rounds, the difference between SAMOSA and other approaches is less evident. For example, after the eighth round, the data points queried using both SAMOSA-L and lfOSA lie close to the model's decision boundaries (as shown in Figure 14). However, some of these decision boundaries may not be the most informative. The bottom cluster in Figure 14a and the the top cluster in Figure 14b are two examples. Here, the selected samples are close to the cluster's boundary but they hardly border with any other cluster. On the other hand, the data points queried using SAMOSA do not fall close to the decision boundaries. This is probably due to the skewed nature of the data points distribution in natural datasets. After a certain amount of AL sampling rounds, the atypical data points lying close to the decision boundaries have already been selected by SAMOSA during the previous iterations. Therefore, it is only possible for SAMOSA to select relevant samples belonging to the known classes, thus explaining the SAMOSA samples falling inside the embedding clusters.

### A.12.5 MORE EXPERIMENTS ON PRECISION VS. EFFECTIVENESS

To further study the *precision* vs. *effectiveness* trade-off, we consider running a custom OSAL sampling procedure on the CIFAR100 dataset in which we artificially set the precision to 1 – i.e. perfect precision – via selecting only samples from the known classes. We sample typical data points only by querying the ones characterized by the lowest memorization scores. We compare the performance of this custom made sampling procedure with an alternative sampling procedure in which we randomly sample data points given a fixed precision rate. Figure 15 presents the results over 30% mismatch ratio. We note that the higher precision selection does not lead to the best performance. In fact, it is almost equivalent to a random sampling with 0.1 less precision in terms of obtained accuracy at the end of the AL procedure. The obtained results also show that small precision variations may not impact the model performance, thus showcasing the importance of sample relevance – i.e., their atypicality – over the precision or recall of the selected samples. We observe a similar trend throughout our experiments on multiple datasets.

### A.12.6 MORE EXPERIMENTS ON SAMOSA-L: PRIORITIZING RECALL

Here, we compare the performance achieved when leveraging SAMOSA-L, which prioritizes sampling precision, against both SAMOSA and the baselines. We present the obtained accuracy in Figure 16 and Tables 7 to 9. Meanwhile, Figures 17 and 18 presents the achieved sampling precision and recall respectively.

After ten rounds, SAMOSA-L represents the second best approach (with rank 2.5), proving the flexibility of relying on SAMIS for selecting data points to be labelled in AL. However, for earlier rounds SAMOSA-L is outperformed by EOAL and lfOSA. In fact, SAMOSA-L becomes the second best selection method after round number eight. This is because SAMOSA-L focuses on selecting clean samples to ensure their belonging to known classes. This means that *(i)* the selected samples

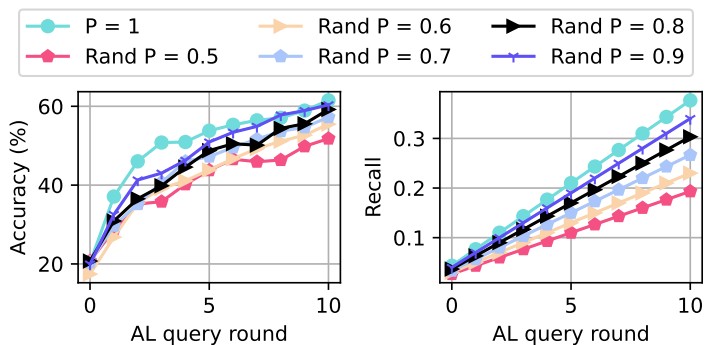

Figure 15: Perfect precision sampling against random sampling with fixed precision over CIFAR100 with 30% mismatch. Perfect precision is as effective as random sampling with 0.1 less precision.

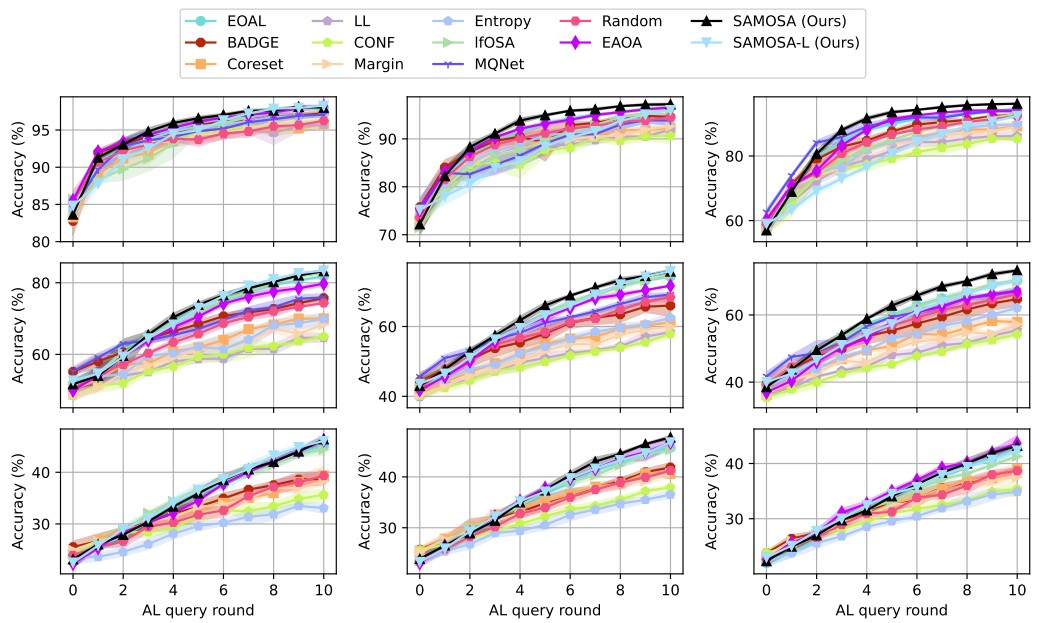

Figure 16: Classification performance comparison on CIFAR10 (first row), CIFAR100 (second row) and Tiny-Imagenet (third row) with 20% (first column), 30% (second column) and 40% (third column) mismatch ratio. Due to resource and time constraints, for the TinyImageNet dataset, we compare SAMOSA with the tractably fast baselines, since the other baselines such as BADGE, MQNet, etc. did not complete the optimization process even after 10 GPU days.

belong to areas of the embedding space which are far from the decision boundaries (as shown in Appendix A.12.4), incurring into small information for model updates; and *(ii)* given the skewed distribution of data points (unbalanced towards clean and easy to classify samples), during the first few rounds SAMOSA-L selects similar data points that bring redundant information. Only whenever the number of rounds increases the variety of the selected samples is sufficient to produce meaningful model updates. Which we verify in Appendix A.12.3.

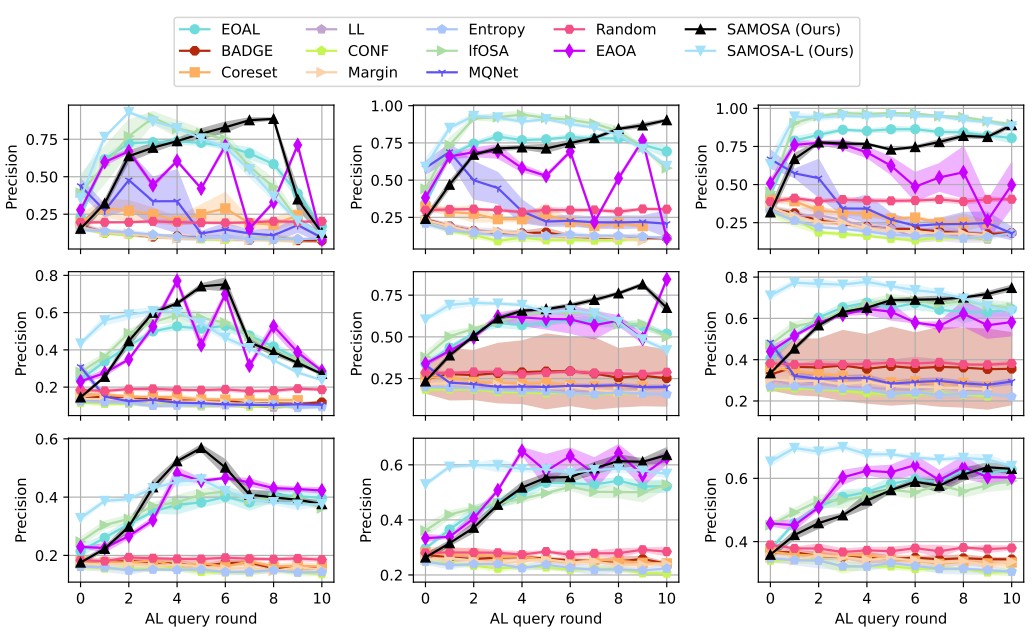

Figure 17: Sampling precision comparison on CIFAR10 (first row), CIFAR100 (second row) and Tiny-Imagenet (third row) with 20% (first column), 30% (second column) and 40% (third column) mismatch ratio.

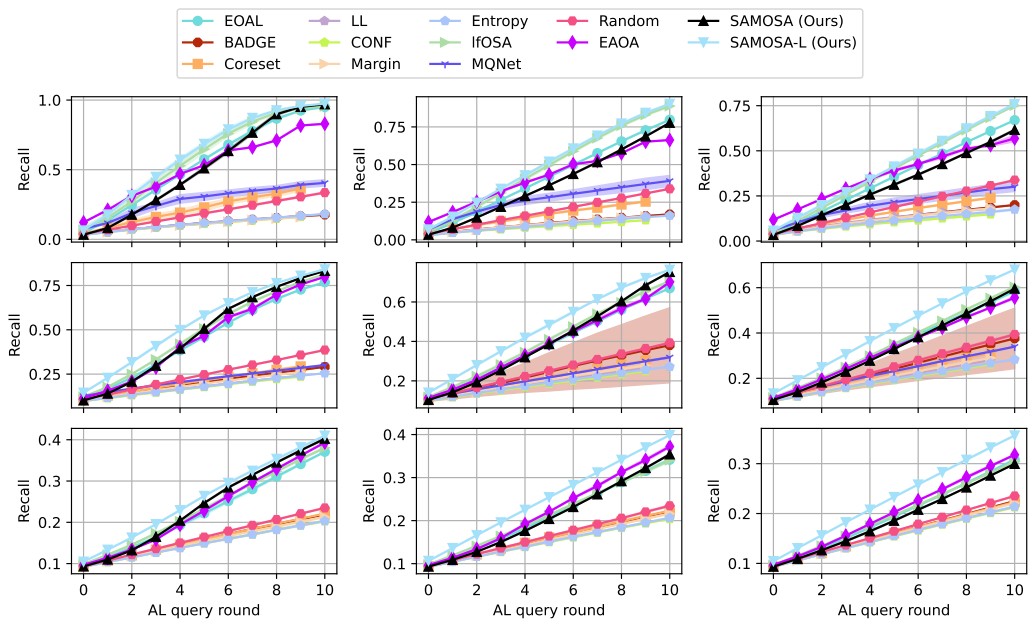

Figure 18: Sampling recall comparison on CIFAR10 (first row), CIFAR100 (second row) and Tiny-Imagenet (third row) with 20% (first column), 30% (second column) and 40% (third column) mismatch ratio.

| Strategy | CIFAR10 | | | CIFAR10 | | | TinyIamgeNet | | | Average Rank (↓) |
|---|---|---|---|---|---|---|---|---|---|---|
| | 20% | 30% | 40% | 20% | 30% | 40% | 20% | 30% | 40% | |
| Random | 96.25 ± 0.15 | 94.50 ± 0.15 | 92.33 ± 0.21 | 74.36 ± 0.38 | 68.40 ± 1.15 | 66.18 ± 0.06 | 39.34 ± 0.85 | 41.23 ± 0.72 | 38.68 ± 0.48 | 7.67 |
| BADGE | 97.26 ± 0.39 | 94.58 ± 0.62 | 92.76 ± 0.40 | 75.74 ± 0.46 | 66.00 ± 3.09 | 64.73 ± 2.15 | 38.73 ± 1.13 | 41.87 ± 0.89 | 38.80 ± 0.61 | 6.89 |
| LL | 95.76 ± 0.62 | 91.92 ± 0.86 | 86.16 ± 1.70 | 64.64 ± 1.18 | 58.37 ± 0.51 | 55.46 ± 1.38 | - | - | - | 12.33 |
| Coreset | 95.78 ± 0.30 | 92.22 ± 0.54 | 89.21 ± 0.70 | 70.00 ± 0.42 | 61.18 ± 1.01 | 57.88 ± 0.89 | 39.39 ± 1.73 | 40.62 ± 0.64 | 38.95 ± 0.89 | 9.22 |
| Entropy | 96.91 ± 0.20 | 94.03 ± 1.04 | 89.76 ± 0.97 | 69.89 ± 1.76 | 62.52 ± 1.25 | 62.13 ± 1.61 | 33.08 ± 0.99 | 36.53 ± 0.77 | 34.90 ± 0.40 | 9.56 |
| CONF | 96.24 ± 0.54 | 90.64 ± 1.61 | 85.35 ± 0.30 | 65.03 ± 0.86 | 57.93 ± 1.12 | 54.25 ± 0.51 | 35.64 ± 1.05 | 37.94 ± 0.53 | 35.38 ± 1.40 | 11.67 |
| Margin | 96.49 ± 0.67 | 92.62 ± 0.44 | 89.09 ± 0.87 | 68.36 ± 0.67 | 59.93 ± 1.11 | 57.20 ± 1.33 | 38.25 ± 0.61 | 40.70 ± 0.72 | 37.55 ± 0.65 | 9.89 |
| MQNet | 97.05 ± 0.22 | 93.82 ± 0.87 | 93.89 ± 0.28 | 75.94 ± 0.93 | 69.19 ± 0.52 | 67.38 ± 1.02 | - | - | - | 6.17 |
| IfOSA | 98.09 ± 0.22 | 96.17 ± 0.19 | 93.07 ± 0.65 | 82.94 ± 0.50 | 74.77 ± 0.42 | 70.00 ± 0.37 | 44.41 ± 1.10 | 45.35 ± 0.15 | 41.29 ± 0.43 | 4.11 |
| EOAL | 97.89 ± 0.27 | 96.49 ± 0.33 | 94.61 ± 0.25 | 81.78 ± 0.52 | 74.29 ± 0.37 | 70.25 ± 0.55 | 45.03 ± 0.84 | 45.76 ± 0.75 | 42.76 ± 0.71 | 3.56 |
| EAOA | 98.23 ± 0.07 | 96.56 ± 0.23 | 94.08 ± 0.18 | 79.75 ± 0.67 | 71.69 ± 0.26 | 67.03 ± 0.62 | 46.18 ± 0.11 | 46.75 ± 0.86 | 43.68 ± 1.39 | 3.22 |
| SAMOSA-L (Ours) | 98.28 ± 0.30 | 96.00 ± 0.58 | 92.43 ± 0.32 | 83.41 ± 0.16 | 75.85 ± 0.33 | 70.29 ± 0.42 | 46.06 ± 0.78 | 46.80 ± 0.46 | 42.36 ± 0.48 | 2.89 |
| SAMOSA (Ours) | 98.00 ± 0.15 | 97.23 ± 0.18 | 96.19 ± 0.08 | 83.24 ± 0.47 | 75.75 ± 0.58 | 73.24 ± 0.30 | 46.38 ± 0.72 | 47.79 ± 0.34 | 43.22 ± 0.32 | 1.67 |

Table 7: Average accuracy of $f_{test}$ after the last active learning rounds across each dataset and mismatch ratio setting. The table presents the average accuracy and its standard deviation over four experiment runs. The best (second-best) approach for each setting is highlighted in green (blue). Additionally, the average rank of each method is reported, calculated as the average placement of the method against others for each experimental setting. SAMOSA achieves the highest accuracy over most settings and has the best average rank. Meanwhile, SAMOSA-L is the second-best approach overall.

| Strategy | CIFAR10 | | | CIFAR10 | | | TinyIamgeNet | | | Average Rank (↓) |
|---|---|---|---|---|---|---|---|---|---|---|
| | 20% | 30% | 40% | 20% | 30% | 40% | 20% | 30% | 40% | |
| Random | 95.49 ± 0.22 | 93.62 ± 0.23 | 90.55 ± 0.57 | 72.05 ± 1.41 | 64.85 ± 0.31 | 62.94 ± 0.33 | 37.16 ± 1.26 | 38.84 ± 1.22 | 35.99 ± 1.38 | 7.56 |
| BADGE | 96.80 ± 0.27 | 94.25 ± 0.64 | 91.18 ± 0.23 | 72.70 ± 0.79 | 63.47 ± 2.22 | 61.64 ± 2.69 | 37.51 ± 0.77 | 39.12 ± 0.63 | 36.62 ± 0.53 | 6.44 |
| LL | 94.64 ± 1.58 | 90.50 ± 0.44 | 85.46 ± 1.97 | 61.53 ± 1.47 | 54.37 ± 0.87 | 51.66 ± 1.15 | - | - | - | 12.33 |
| Coreset | 94.88 ± 0.52 | 91.77 ± 0.20 | 88.16 ± 0.38 | 68.31 ± 0.82 | 59.43 ± 0.76 | 56.16 ± 1.52 | 35.96 ± 0.32 | 38.69 ± 0.70 | 36.92 ± 0.32 | 9.33 |
| Entropy | 96.79 ± 0.35 | 92.87 ± 0.84 | 88.68 ± 0.39 | 68.14 ± 1.71 | 59.67 ± 1.31 | 58.11 ± 0.30 | 31.88 ± 1.22 | 34.73 ± 1.10 | 33.19 ± 1.26 | 9.56 |
| CONF | 95.48 ± 0.95 | 89.60 ± 0.81 | 83.73 ± 1.30 | 62.30 ± 0.66 | 53.94 ± 0.63 | 51.40 ± 0.99 | 33.45 ± 0.73 | 35.52 ± 0.72 | 33.29 ± 0.38 | 11.67 |
| Margin | 95.96 ± 0.48 | 90.70 ± 1.36 | 87.12 ± 0.37 | 67.20 ± 1.23 | 57.74 ± 0.90 | 54.87 ± 1.32 | 36.91 ± 0.47 | 38.57 ± 0.52 | 35.55 ± 0.83 | 10.00 |
| MQNet | 96.45 ± 0.32 | 92.99 ± 0.89 | 93.10 ± 0.41 | 73.15 ± 1.75 | 66.43 ± 0.70 | 65.11 ± 0.73 | - | - | - | 6.00 |
| IfOSA | 97.45 ± 0.24 | 94.52 ± 0.47 | 90.58 ± 0.51 | 79.94 ± 0.68 | 71.35 ± 0.52 | 66.79 ± 0.94 | 41.51 ± 0.93 | 42.37 ± 0.53 | 38.19 ± 0.15 | 4.11 |
| EOAL | 97.28 ± 0.26 | 95.52 ± 0.06 | 93.03 ± 0.13 | 79.35 ± 0.23 | 71.81 ± 0.20 | 66.70 ± 0.59 | 41.96 ± 1.10 | 42.88 ± 0.88 | 40.03 ± 0.34 | 3.56 |
| EAOA | 97.38 ± 0.28 | 95.60 ± 0.21 | 94.05 ± 0.35 | 77.56 ± 0.98 | 69.12 ± 0.72 | 64.93 ± 0.17 | 42.05 ± 0.59 | 43.61 ± 0.33 | 40.19 ± 0.13 | 3.22 |
| SAMOSA-L (Ours) | 97.86 ± 0.21 | 93.74 ± 0.71 | 89.28 ± 0.19 | 81.03 ± 0.16 | 72.07 ± 0.23 | 66.49 ± 0.44 | 43.26 ± 0.56 | 43.29 ± 0.73 | 39.36 ± 0.29 | 3.33 |
| SAMOSA (Ours) | 97.61 ± 0.29 | 96.83 ± 0.07 | 95.73 ± 0.19 | 80.20 ± 0.06 | 73.28 ± 0.68 | 70.06 ± 0.31 | 42.01 ± 0.44 | 44.55 ± 0.32 | 39.99 ± 0.89 | 1.67 |

Table 8: Average accuracy of $f_{test}$ after *eight* active learning rounds across each dataset and mismatch ratio setting. The table presents the average accuracy, its standard deviation and the average rank of each method. The best (second-best) approach for each setting is highlighted in green (blue). SAMOSA achieves the highest accuracy over most settings and has the best average rank. Moreover, SAMOSA achieves either the highest or the second highest accuracy for all settings. Meanwhile, SAMOSA-L is the second-best approach overall. However, in this case SAMOSA-L ties with EOAL for second place (according to the average rank scoring system).

| Strategy | CIFAR10 | | | CIFAR10 | | | TinyIamgeNet | | | Average Rank (↓) |
|---|---|---|---|---|---|---|---|---|---|---|
| | 20% | 30% | 40% | 20% | 30% | 40% | 20% | 30% | 40% | |
| Random | 93.71 ± 0.87 | 91.19 ± 0.59 | 86.67 ± 0.29 | 65.71 ± 1.01 | 58.38 ± 1.71 | 56.89 ± 0.46 | 31.78 ± 0.95 | 33.95 ± 0.56 | 31.26 ± 0.57 | 8.00 |
| BADGE | 95.66 ± 0.56 | 89.18 ± 3.80 | 87.63 ± 0.45 | 68.45 ± 1.20 | 57.96 ± 2.28 | 55.29 ± 2.24 | 33.33 ± 1.01 | 34.90 ± 1.32 | 32.58 ± 0.89 | 6.56 |
| LL | 94.03 ± 0.72 | 88.53 ± 0.51 | 81.64 ± 1.01 | 58.74 ± 1.21 | 50.99 ± 1.57 | 48.06 ± 0.72 | - | - | - | 11.83 |
| Coreset | 94.26 ± 0.65 | 89.43 ± 0.46 | 84.26 ± 0.43 | 61.94 ± 1.55 | 55.00 ± 0.90 | 50.79 ± 1.11 | 33.01 ± 0.97 | 35.18 ± 1.22 | 32.79 ± 1.14 | 8.44 |
| Entropy | 95.50 ± 0.22 | 90.77 ± 0.29 | 84.48 ± 0.97 | 62.13 ± 0.94 | 54.27 ± 0.94 | 53.08 ± 0.38 | 29.58 ± 0.88 | 30.56 ± 0.96 | 29.58 ± 0.65 | 8.78 |
| CONF | 94.46 ± 0.34 | 87.70 ± 1.17 | 79.13 ± 1.01 | 61.26 ± 1.17 | 53.00 ± 0.90 | 49.64 ± 0.58 | 31.81 ± 1.82 | 32.22 ± 0.38 | 31.02 ± 0.23 | 11.33 |
| Margin | 94.31 ± 0.80 | 89.25 ± 0.48 | 84.03 ± 1.35 | 61.26 ± 1.17 | 53.00 ± 0.90 | 49.64 ± 0.58 | 33.31 ± 0.77 | 33.79 ± 1.05 | 32.57 ± 0.54 | 9.44 |
| MQNet | 94.86 ± 0.65 | 89.03 ± 2.33 | 90.86 ± 0.45 | 67.08 ± 0.85 | 61.04 ± 1.20 | 59.41 ± 0.45 | - | - | - | 6.33 |
| IfOSA | 95.21 ± 0.74 | 89.74 ± 0.81 | 85.04 ± 0.36 | 71.96 ± 0.83 | 63.47 ± 0.69 | 59.69 ± 0.40 | 36.70 ± 0.58 | 37.32 ± 0.32 | 33.24 ± 0.78 | 4.22 |
| EOAL | 95.63 ± 0.36 | 93.27 ± 0.38 | 89.96 ± 0.64 | 72.50 ± 1.40 | 63.87 ± 0.73 | 59.95 ± 0.54 | 35.99 ± 0.74 | 36.67 ± 0.52 | 34.24 ± 0.83 | 3.11 |
| EAOA | 96.14 ± 0.42 | 93.18 ± 0.21 | 91.44 ± 0.74 | 70.55 ± 0.53 | 62.62 ± 1.33 | 57.88 ± 0.26 | 34.53 ± 0.25 | 37.82 ± 0.27 | 34.99 ± 0.44 | 3.22 |
| SAMOSA-L (Ours) | 94.95 ± 0.20 | 88.43 ± 0.63 | 80.85 ± 0.93 | 73.00 ± 0.54 | 63.20 ± 0.61 | 57.24 ± 1.24 | 36.55 ± 1.19 | 37.13 ± 1.05 | 34.50 ± 0.31 | 5.67 |
| SAMOSA (Ours) | 96.59 ± 0.24 | 94.93 ± 0.25 | 93.56 ± 0.41 | 73.83 ± 0.77 | 65.99 ± 0.87 | 62.77 ± 0.98 | 35.91 ± 0.76 | 37.35 ± 0.83 | 34.06 ± 0.71 | 1.78 |

Table 9: Average accuracy of $f_{test}$ after *five* active learning rounds across each dataset and mismatch ratio setting. The table presents the average accuracy, its standard deviation and the average rank of each method. The best (second-best) approach for each setting is highlighted in green (blue). SAMOSA achieves the highest accuracy over most settings and has the best average rank. Meanwhile, SAMOSA-L does not represent a very successful approach ranking only fourth amongst the compared approaches (average rank of 5). The reasons behind this SAMOSA-L behaviour are hypothesized and tested in Appendices A.12.3 and A.12.4.

