# OpenReview forum: "SAMOSA: Sharpness Aware Minimization for Open Set Active learning"
_ICLR.cc/2026/Conference — Submitted to ICLR 2026_

### Official Review · Reviewer_T42G · 2025-10-22

**Soundness:** 2
**Presentation:** 1
**Contribution:** 2
**Rating:** 2
**Confidence:** 5

**Summary:**

This paper introduces SAMOSA, an inconsistency-based querying strategy that leverages the prediction discrepancy between models trained with SGD and Sharpness-Aware Minimization (SAM). The inconsistency measure, termed SAMIS-P, is defined as the $L_1$ difference between the output probabilities of the two models. SAMOSA adopts a two-step querying framework: first, both models act as $(K+1)$-class classifiers, where the additional class helps filter out potential unknown samples; second, samples with higher SAMIS-P scores are prioritized for labeling. Extensive experiments on CIFAR10, CIFAR100, and TinyImageNet show that the proposed method performs well.

**Strengths:**

1. This paper attempts to provide a new perspective on open-set active learning from a theoretical point of view.
2. The paper provides high-quality visualization results.

**Weaknesses:**

1. The entropy-based experiments in this paper seem somewhat problematic. Normally, in active learning, the model tends to select samples with higher entropy. However, as shown in Figure 3, the selected samples actually have lower entropy values. Similarly, in Figure 1(c), the LfOSA method is not uncertainty-based; it instead favors samples that are most likely to belong to known classes, i.e., it is purity-based. In Figure 1(b), the samples visualized by the authors also do not appear to correspond to those with high entropy values.

2. The authors essentially adopt a typical inconsistency-based sampling strategy commonly used in active learning. What is relatively new is their emphasis on training models using SAM. However, it is unclear what the particular advantage of SAM is in this context—it seems mainly used to increase the prediction discrepancy between the two models by altering the training procedure.

3. A key challenge in open-set active learning is how to distinguish hard samples within known classes from unknown-class samples. In this regard, the paper does not provide any new insights.

4. The authors only include EOAL as the most recent baseline, lacking comparisons with other newer state-of-the-art methods. Moreover, the reported results seem somewhat questionable — in Table 1, EOAL performs worse than LfOSA in several cases, which raises some doubts.

5. In Figure 4, the curves of the compared methods are not clearly presented; in some cases, certain baselines even appear to perform better, but the visualization makes this unclear.

**Questions:**

Please see the Weaknesses.

---

> ### Author Response · Authors · 2025-11-21
>
> We thank the reviewer for providing valuable feedback and comments.
>
> **Figure 3 and low entropy**:
> We clarify that Figure 3 focuses exclusively on the **misclassified** samples queried for active learning. The red area shows samples that are queried by entropy and the blue area shows samples queried by SAMOSA. Therefore, Figure 3 clearly shows that SAMOSA queries samples the model was confidently wrong on (low entropy in x axis) while entropy based selection misses out on them.
>
> **Confusion about Figure 1(c)**:
> Thank you for pointing this out. We acknowledge that our description in the paper incorrectly referred to Figure 1(c) as illustrating uncertainty-based sampling behavior, whereas the figure actually visualizes the sampling pattern of lfOSA, which is purity-based. This was an oversight in wording, and we appreciate the reviewer for catching it.
>
> **Purpose of Figure 1(b)**:
> We clarify that Figure 1(b) was intended to serve as a baseline comparison illustrating what typical examples look like when purity is prioritized over informativeness. The goal was to contrast these common samples with the highly informative and relevant examples that entropy-based or uncertainty-driven approaches fail to select because the model confidently misclassifies them.
> This design choice highlights a key limitation of purity-focused strategies: selecting only samples with very high purity scores tends to yield typical, redundant examples that add little new information to the model. In contrast, SAMOSA deliberately targets samples near decision boundaries that drive meaningful improvements in generalization.
>
> **On SAMOSA as inconsistency-based sampling**:
> We respectfully disagree with the characterization that SAMOSA merely adopts a typical inconsistency-based sampling strategy. While SAMOSA leverages prediction discrepancy, its novelty lies in (i) the theoretical foundation for why this discrepancy correlates with sample atypicality and informativeness, and (ii) the use of Sharpness-Aware Minimization (SAM) as a principled mechanism not just to “increase discrepancy,” but to capture samples that standard SGD fails to generalize well (Theorem 2.1). SAM’s role is crucial because:
>
> * It amplifies the signal for hard ID samples and informative OOD samples that lie near decision boundaries (where SGD and SAM disagree).
> * It enables SAMOSA to identify samples that are wrongly classified with high confidence. These samples are overlooked by entropy-based or margin-based methods while SAMIS-P can capture them.
> * It improves robustness to label noise (Table 2), a property rooted in SAM’s optimization dynamics.
>
> **OOD filtering**:
> We do not believe that distinguishing hard samples within known classes (hard IDs) from unknown-class samples (OODs) is the most relevant aspect of OSAL. [1] demonstrates that informativeness is the key driver of performance. Their findings show that sampling informative OOD samples can significantly improve OSAL performance. Hard ID samples are indeed important, but informative OOD samples lying near decision boundaries improve the model’s ability to separate relevant and irrelevant classes in later rounds.
> The experiments we provide in Section 7 strongly support this insight:
>
> * SAMOSA selects more OOD samples (lower precision) than lfOSA and EOAL, yet achieves higher accuracy, showing that informativeness outweighs purity (Figures 5 & 7).
> * SAMOSA-L (favoring purity) yields the highest precision but fails to achieve top performance, confirming that ID/OOD separation alone does not guarantee effectiveness (Figure 18).
> * SAMOSA consistently outperforms baselines across datasets and mismatch ratios, despite lower early-round precision (Table 1 & Figure 4).
>
> **Baselines**:
> We included EOAL and lfOSA as the two most recent OSAL methods for which reproducible code was available at the time of implementing our experiments. Both approaches were published very recently and represent the latest advancements in OSAL. We are fully open to extending our experimental evaluation if the reviewer can share additional reproducible implementations of newer methods.
>
> Regarding the observation that EOAL performs worse than lfOSA in several cases, this is not correct. Across all our reported tables,  EOAL consistently performs better than lfOSA. In fact, EOAL is the second-best approach overall, with an average rank of 2.3.
>
> **Figure 4 and SAMOSA performance**:
> We acknowledge that Figure 4 may appear crowded due to the large number of baselines compared (10 methods plus SAMOSA). This was intentional to provide a comprehensive view. However, Figure 4 is complemented by Table 1 in the main text and Tables 5 and 6 in the Appendix, which provide detailed numerical results for multiple AL rounds.
>
> We kindly hope the reviewer may consider our response when revising their scores.
>
> [1]. Yang Yang, Yuxuan Zhang, Xin Song, and Yi Xu. Not all out-of-distribution data are harmful to openset active learning. NeurIPS 2023

---

> ### Comment · Reviewer_T42G · 2025-11-24
>
> Thank you for the clarification. I acknowledge that my previous comment on Figure 3 was mistaken, and several of my other concerns have also been partially addressed. Therefore, I have raised my overall score to 4.
>
> My remaining concerns are as follows. The proposed method essentially adopts a standard inconsistency-based sampling strategy in active learning. To make such inconsistency-based criteria effective, it is often necessary to employ different training schemes or modes; in this sense, the contribution appears incremental rather than a substantially novel advance. Moreover, the authors do not provide experiments to demonstrate that their strategy is more effective than other existing inconsistency-based strategies. I am also concerned about the timeliness of the baselines: the comparison set does not sufficiently cover recent methods, including [1] mentioned by the authors, as well as [2], [3], and [4], among others.
>
> [2] Bidirectional uncertainty-based active learning for open-set annotation. ECCV2024.
>
> [3] Inconsistency-Based Data-Centric Active Open-Set Annotation. AAAI2024.
>
> [4] Rethinking Epistemic and Aleatoric Uncertainty for Active Open-Set Annotation: An Energy-Based Approach. CVPR2025.

---

> ### Author Response · Authors · 2025-12-03
>
> We thank the reviewer for their valuable feedback and constructive comments, as well as for acknowledging the quality of our rebuttal and increasing the review score. We would like to provide further clarifications regarding the remaining concerns:
>
> **On SAMOSA as a standard inconsistency-based sampling strategy**: We partially agree that SAMOSA leverages an inconsistency-based approach for sampling informative data points. However, we respectfully disagree that this is the sole reason for its performance. As shown in Section 3.3 and Appendix A.9 of our original submission, the gains achieved by SAMOSA are not merely due to an ensemble effect or the discrepancy between multiple neural networks. Specifically, we compare SAMOSA against a disagreement-based sampling strategy using three SGD models, following Algorithm 0 in Hanneke (2014) [1] combined with K+1 filtration (see Table 3). While this baseline is effective, it consistently underperforms compared to SAMOSA. This demonstrates that SAMOSA’s advantage stems from its unique design which is also our main algorithmic novelty, particularly the performance gap between SAM and SGD on atypical samples. Further, our algorithm is firmly grounded in novel theoretical formulation intuiting why the sam vs sgd atypicality differential is helpful as formalized in Theorem 3.1.
>
> **On the comparison with recent approaches**: We appreciate the reviewer for pointing out additional baselines [2], [3], and [4]. After receiving the comment, we implemented and tested the most recent and competitive baseline EAOA [4] (CVPR 2025) which has been proven to be superior to [2,3] as per [4], under the same experimental setup described in Sections 5 and A.11.
> Below are the results comparing SAMOSA and EAOA across CIFAR10, CIFAR100, and TinyImageNet at varying mismatch ratios:
>
> | AL round | Approach | CIFAR10 20% | CIFAR10 30% | CIFAR10 40% | CIFAR100 20% |CIFAR100 30% | CIFAR100 40% | TinyImagenet 20% | TinyImagenet 30% | TinyImagenet 40% | Avg. Rank |
> |-------|-------|-------|-------|-------|-------|-------|-------|-------|-------|-------|-------|
> 4 | **EAOA**  | 95.33 | 92.13 | 88.14 | 67.59 | 59.86 | 53.03 | 32.10 | **35.24** | **32.72** | 2.89 |
> 4 | **SAMOSA (Ours)**  | **95.95** | **93.82** | **91.63** | **70.35** | **61.78** | **58.88** | **33.35** | 35.00 | 31.52 | **1.78** |
>   |  |  |  |  |  |  |  |  |  |  |  |
> 6 | **EAOA**  | 96.68 | 94.00 | 92.61 | 74.16 | 65.29 | 60.94 | 37.79 | 39.65 | **37.01** | 2.78 |
> 6 | **SAMOSA (Ours)**  | **97.03** | **95.88** | **94.29** | **76.51** | **68.82** | **65.79** | **38.50** | **40.39** | 36.21 | **1.22** |
>   |  |  |  |  |  |  |  |  |  |  |  |
> 8 | **EAOA**  | 97.38 | 95.60 | 94.05 | 77.56 | 69.12 | 64.93 | **42.05** | 43.61 | **40.19** | 2.67 |
> 8 | **SAMOSA (Ours)**  | **97.61** | **96.83** | **95.73** | **80.20** | **73.28** | **70.06** | 42.01 | **44.55** | 39.99 | **1.33** |
>   |  |  |  |  |  |  |  |  |  |  |  |
> 10 | **EAOA**  | **98.23** | 96.56 | 94.08 | 79.75 | 71.69 | 67.03 | 46.18 | 46.75 | **43.68** | 2.67 |
> 10 | **SAMOSA (Ours)**  | 98.00 | **97.23** | **96.19** | **83.24** | **75.75** | **73.24** | **46.38** | **47.79** | 43.22 | **1.33** |
>
>
> SAMOSA consistently outperforms EAOA across all datasets and settings, except for TinyImageNet at 40% mismatch ratio and CIFAR10 at 20% mismatch ratio where performance is comparable. Overall, SAMOSA achieves up to 4% higher accuracy in specific configurations, confirming its state-of-the-art performance. We have updated the submission PDF to include these results and thank the reviewer for highlighting these baselines.
>
> [1] Steve Hanneke. "Theory of disagreement-based active learning." Foundations and Trends in Machine Learning, 2014
>
> [2] Mao, Ruiyu et al. "Inconsistency-based data-centric active open-set annotation." AAAI 2024.
>
> [3] Zong, Chen-Chen et al. “Bidirectional uncertainty-based active learning for open-set annotation.” ECCV 2024.
>
> [4] Zong, Chen-Chen et al. “Rethinking Epistemic and Aleatoric Uncertainty for Active Open-Set Annotation: An Energy-Based Approach.” CVPR 2025.

---

### Official Review · Reviewer_B1Uz · 2025-10-30

**Soundness:** 3
**Presentation:** 3
**Contribution:** 3
**Rating:** 6
**Confidence:** 4

**Summary:**

This paper proposes SAMOSA, a method for open-set active learning that leverages the difference in predicted probabilities between a model trained with SGD and one trained with sharpness-aware minimization (SAM). This discrepancy, called SAMIS-P, is proposed as a proxy for data atypicality. The approach iteratively selects unlabeled samples with the highest SAMIS-P scores for labeling. Extensive experiments on CIFAR10, CIFAR100, and TinyImageNet show modest accuracy improvements over several baselines.

**Strengths:**

S1. **Clear problem motivation.** The paper correctly identifies the limitation of entropy-based sampling under open-set settings, where uncertainty measures become unreliable due to distributional shift.

S2. **Solid experimental setup.** Multiple datasets (CIFAR-10, CIFAR-100, TinyImageNet) and several baselines are used. The results are consistently positive, suggesting that the proposed typicality metric is at least empirically stable.

S3. **Readable presentation.** The figures and method overview are clear, and the paper is well written overall. It is easy to understand the intuition and pipeline of the approach.

**Weaknesses:**

W1. **Limited novelty in methodology.** The approach primarily combines two existing techniques, SAM optimization and active learning, in a quiet straightforward way. The “typicality discrepancy” is a small conceptual extension rather than a fundamentally new algorithmic idea. The authors should more emphasize why SAM optimization has a large synergy when applied to open set active learning domain.

**Questions:**

I have no question.

---

> ### Author Response · Authors · 2025-11-21
>
> We thank the reviewer for taking the time to read our paper and providing valuable feedback.
>
> **On the limited novelty**:
> While our approach indeed leverages SAM and active learning, we respectfully believe that considering it as a straightforward combination understates the novelty and significance of our contributions. We here try to clarify briefly the relevant novelties of our paper:
>
> * To our knowledge, the link between loss sharpness and atypical examples has not been firmly established in OSAL literature. Previous works have only hinted at this connection experimentally (e.g., [1]). Our work is the first to formalize this relationship through Theorem 2.1, showing that SAM and SGD exhibit similar generalization on typical samples but differ significantly on atypical ones. This theoretical insight is non-trivial and forms the basis for our sampling criterion.
> * We introduce a novel metric for informativeness sample selection in OSAL, SAMIS-P, a principled metric based on the discrepancy between SAM and SGD predictions. Unlike entropy or confidence-based methods, SAMIS-P captures samples that are confidently misclassified—examples that traditional OSAL approaches systematically ignore. This mechanism directly addresses a key limitation of existing methods and represents a conceptual advance beyond uncertainty-based sampling.
> * We provide a new perspective on multi-round dynamics of OSAL: our findings in Section 7 reveal a novel trade-off, which is prioritizing informativeness may reduce sample purity in early rounds, but this effect is compensated later as the model learns to optimize decision boundaries. This dynamic leads to higher precision and superior accuracy in later rounds (Figures 7 and 18), offering a fresh perspective on OSAL beyond static purity/informativeness trade-offs.
> * We provide the first disagreement-based approach grounded in loss geometry for OSAL. While OSAL extends from classical AL, most existing methods adapt uncertainty or clustering strategies. To the best of our knowledge, SAMOSA is the first OSAL method to incorporate a disagreement-based criterion rooted in loss sharpness, resulting in state-of-the-art performance across datasets and mismatch ratios (Table 1).
> * We find that SAM’s ability to learn flatter minima amplifies signals for hard IDs and informative OODs near decision boundaries and show how this synergy is particularly impactful in OSAL, where distinguishing relevant from irrelevant classes is challenging. By exploiting this property, SAMOSA improves both generalization and robustness to label noise (see Table 2), which is critical in real-world annotation scenarios.
>
> We hope these clarifications help the reviewer better appreciate the novelty and contributions of our work. We kindly hope the reviewers may consider this when revising their scores.
>
> [1] Approximating Memorization Using Loss Surface Geometry for Dataset Pruning and Summarization. KDD 2024.

---

> > ### Comment · Reviewer_B1Uz · 2025-11-26
> > **Response by Reviewer B1Uz**
> >
> > I appreciate the authors' detailed feedback regarding my concerns. I have decided to retain my score.

---

### Official Review · Reviewer_DZBW · 2025-11-04

**Soundness:** 2
**Presentation:** 3
**Contribution:** 3
**Rating:** 6
**Confidence:** 5

**Summary:**

The paper introduces SAMOSA, an open-set active learning (OSAL) method that scores unlabeled samples by the probability gap between SAM and SGD and filters unknowns with a K+1 head, prioritizing atypical, boundary-proximal examples. It is theoretically motivated that SGD–SAM predictions diverge on atypical samples but not on typical ones, and this is validated via embedding analyses across training. It outperforms many OSAL baselines by up to 2–3% on CIFAR/Tiny-Imagenet datasets with comparable runtime.

**Strengths:**

- Theory-to-practice link: a simple, label-free atypicality proxy (|pSAM−pSGD|) that correlates with decision-boundary informativeness.
- Empirical strength: multi-dataset gains, boundary visualizations, ablations (OOD selection value, ensemble vs. SAM effect), and robustness to label noise.
- Practicality and efficiency: easy to plug in (two standard models with a K+1 head) and competitive runtime comparable to strong OSAL baselines.

**Weaknesses:**

Unclear Motivation

- It is unclear whether Fig 2 and 3 convincingly demonstrate that SAMOSA is a better OSAL metric compared to entropy. A good open-set AL metric should effectively filter out OOD samples while selecting informative in-distribution (IN) samples. Showing the proportion of true IN vs. OOD samples among the selected points would better highlight SAMOSA’s efficacy.
- Even after carefully reading the Introduction and Section 2, the reviewer remains unclear about the high-level intuition behind why the L1 norm between ( f_{SAM}(x) ) and ( f_{SGD}(x) ) leads to better OSAL sample selection.

Questionable Experimental Setup

- The experiments rely solely on synthetic datasets derived from CIFAR-10/100 and Tiny-ImageNet (via random class splits). Evaluation on a real open-set noise dataset would provide stronger evidence.
- There are no results for 0% mismatch ratios. Since real-world data collection often occurs in very low-noise settings, it is important to verify whether SAMOSA remains effective under such conditions.

**Questions:**

Open-set noise can arise in various domains beyond vision, such as NLP. It would be interesting to see whether SAMOSA still remains effective in non-vision domains.

---

> ### Author Response · Authors · 2025-11-21
>
> We thank the reviewer for providing valuable feedback and constructive comments.
>
> **OOD filtering**:
> We respectfully note that, as demonstrated in [1], strict ID/OOD separation is not the most fundamental aspect of OSAL. Their analysis shows that informativeness (not purity) is the key driver of performance, and that sampling informative OOD samples can significantly improve OSAL outcomes. Our work corroborates these findings and builds on this intuition by focusing on informativeness rather than enforcing a distinguisher between hard IDs and OODs.
>
> To further address the reviewer’s point, we emphasize that we provide precision scores for all OSAL techniques in Figure 7 of Appendix A.12.2. Precision measures the proportion of selected samples that belong to known classes (ID), thus explicitly target the reviewer's request.
>
> This dynamic confirms our core insight: sampling informative samples early (even if they include OODs) pays off later with higher precision and better overall performance. This trend is consistent across all datasets and mismatch ratios, as shown in Figure 7 and discussed in detail in Appendix A.12.2 and A.12.3.
>
> **L1 norm leading to better OSAL selection**:
> We appreciate the reviewer’s request for more intuition behind why the L1 norm between SAM and SGD leads to better OSAL sample selection. SAMOSA builds on Theorem 2.1 (Section 2), which shows that SAM and SGD exhibit similar generalization on typical samples but differ significantly on atypical samples. This difference reflects SAM’s ability to learn flatter minima, improving robustness and generalization for hard or rare patterns. The SAMIS-P score, defined as the L1 norm between the two prediction vectors, is therefore a theoretically motivated proxy for sample informativeness.
>
> Figures 2, 12, 13, 14 corroborate this intuition showing that the samples with high SAMIS-P scores consistently occupy critical positions near decision boundaries during training, confirming their informativeness.
>
> **On “real” open-set noise**:
> The concept of a “real” open-set noise dataset is somewhat ambiguous, as OSAL inherently involves simulating unknown classes and mismatched distributions. Our experimental setup already resembles realistic open-set conditions by:
> Introducing different mismatch ratios (20%, 30%, 40%) to simulate varying degrees of unknown-class presence.
> Including label noise experiments (Table 2), which reflect real-world annotation imperfections.
> These settings are consistent with established OSAL protocols and aim to capture practical challenges in open-set scenarios.
>
> **On the choice of the datatasets**:
> We rely on CIFAR-10, CIFAR-100, and Tiny-ImageNet because these datasets represent the golden standard for OSAL evaluation in the literature. Virtually all recent OSAL methods, including EOAL, lfOSA, and MQNet, report results on these benchmarks. Using these datasets ensures fair comparison with prior work and allows us to validate SAMOSA under widely accepted, reproducible conditions.
>
> **On the 0% mismatch ratio**:
> A 0% mismatch ratio would imply that there are no known classes (ID) in the unlabeled pool, which means there is nothing to actively learn for the target task. In such a scenario, the OSAL setting becomes ill-defined because the goal of active learning is to improve performance on known classes. Therefore, we believe this setting is not meaningful for OSAL evaluation. We assume the reviewer intended to refer to the opposite extreme, 100% known classes (no open-set noise). This scenario corresponds to classical active learning, where all unlabeled samples belong to the known classes and there is no risk of sampling OOD data. In this case, OSAL-specific mechanisms (such as distinguishing ID vs. OOD) are unnecessary. The only relevant criterion becomes informativeness, which is exactly what SAMOSA optimizes.
> Under this setting, SAMOSA would still work effectively by selecting atypical samples that contain rare patterns, improving generalization. This intuition is widely applied in standard AL literature [3,4] and is supported by prior work leveraging SAM vs. SGD discrepancy for dataset pruning [5], where selecting the most informative samples leads to superior performance.
>
> We kindly hope the reviewers may consider this when revising their scores.
>
>
> [1].  Yang Yang, Yuxuan Zhang, Xin Song, and Yi Xu. Not all out-of-distribution data are harmful to openset active learning. NeurIPS 2023
>
> [2]. Christina Baek, Zico Kolter, and Aditi Raghunathan. Why is sam robust to label noise?. Arxiv 2024
>
> [3] Christen, Victor, et al. Informativeness-based active learning for entity resolution. ECML PKDD, 2019.
>
> [4] Song, Miao-Hui, et al. ProxySampler: Proxy Informativeness Estimation for Efficient Data Selection in Active Learning. CIKM, 2025.
>
> [5] Agiollo, Andrea, et al. Approximating Memorization Using Loss Surface Geometry for Dataset Pruning and Summarization. KDD, 2024.

---

### Author Response · Authors · 2025-12-03
**Final Remarks and Summary of Contributions**

We thank the reviewers for their constructive feedback and for helping us improve the quality of our submission. We believe that we addressed all major concerns and substantially clarified the manuscript. Below, we summarize the key revisions we made and the relevance of our contributions.

**Clarification about OOD filtering and motivations:** We revised the introduction and Section 3 to explain why SAMOSA is effective without explicitly optimizing ID/OOD separation. As hinted in [1] and backed by our findings, strict ID/OOD separation is not the key driver of OSAL performance, rather informativeness is. Sampling informative OOD samples significantly improves OSAL outcomes. SAMOSA operationalizes this principle and captures:
1. hard ID samples that improve classification of known classes and boost final model performance; and
2. informative OOD samples that strengthen the K+1 classifier for future OOD filtration, improving purity over time.

We revised section 3 to emphasize that the precision scores available in Figure 7 explicitly measure the quality of the sampling process. Our results show that, initially, SAMOSA’s precision is slightly lower than the EOAL and LfOSA baselines (prioritizing informativeness over purity), but precision rises sharply in later rounds, achieving the highest precision and accuracy. Therefore, SAMOSA can still effectively learn to separate ID and OOD samples, while achieving state-of-the-art accuracy.


**Clarification about Figure 1:** We corrected the wording oversight in the introduction. However, the core message of Figure 1 remains two-folded:
1. uncertainty-based sampling misses confidently misclassified samples. These samples are extremely informative for improving decision boundaries.
2. Purity-based sampling selects redundant, uninformative examples, limiting generalization.

In contrast, Figure 1(d) demonstrates SAMOSA’s advantage: it focuses on samples near decision boundaries (whether hard IDs or informative OODs) that drive meaningful improvements in generalization.

**Additional comparison with baselines:** Following reviewer T42G’s suggestion, we implemented the most recent and strongest known additional baseline EAOA [2] which is already known to outperform other suggested baselines as per [2] under our experimental setup and included it in the revised version of the paper. SAMOSA outperforms EAOA across most datasets and settings, confirming SAMOSA’s state-of-the-art performance.

**Paper relevance:** We also try to clarify briefly the relevant novelties of our paper:
- To our knowledge, the link between loss landscape geometry (i.e. sharpness) and atypical examples has not been firmly theoretically established. Previous works have only hinted at this connection experimentally (e.g., [3]). Our work is the first to formalize this relationship through Theorem 2.1, showing that SAM and SGD exhibit similar generalization on typical samples but differ significantly on atypical ones. This theoretical insight is non-trivial and forms the basis for our sampling criterion.
- We introduce a novel metric for informativeness sample selection in OSAL, SAMIS-P, a principled metric based on the discrepancy between SAM and SGD predictions. This mechanism directly addresses a key limitation of existing methods and represents a conceptual advance beyond uncertainty- and purity-based sampling.
- We provide a new perspective on multi-round dynamics of OSAL: our findings reveal a novel trade-off, prioritizing informativeness may reduce sample purity in early rounds, but this effect is compensated later as the model learns to optimize decision boundaries.
- We provide the first disagreement-based approach grounded in loss geometry for OSAL, resulting in state-of-the-art performance across datasets and mismatch ratios.
- We find that SAM’s ability to learn flatter minima amplifies signals for hard IDs and informative OODs near decision boundaries and show how this synergy is particularly impactful in OSAL. By exploiting this property, SAMOSA improves both generalization and robustness to label noise, critical in real-world annotation scenarios.

We trust that these clarifications and additions address the reviewers’ concerns and provide the area chairs with a clear understanding of the manuscript’s contributions and impact within the ICLR community.

[1] Yang, Yang, et al. "Not all out-of-distribution data are harmful to open-set active learning." NeurIPS 2023.

[2] Zong, Chen-Chen et al. “Rethinking Epistemic and Aleatoric Uncertainty for Active Open-Set Annotation: An Energy-Based Approach.” CVPR 2025.

[3] Agiollo, Andrea, et al. “Approximating Memorization Using Loss Surface Geometry for Dataset Pruning and Summarization.” KDD 2024.

---

### Author Response · Authors · 2025-12-03
**Summary of Rebuttal Discussion for the AC**

Dear AC,

Given the changes to the review process, we summarize the discussion to assist your decision, aiming to be as objective as possible.

**Reviewer DZBW - initial score: 6**
- Positive opinion overall, highlighting theory-to-practice link, empirical strength, and efficiency.
- Requested clarifications on SAMOSA’s OOD filtering; we provided a detailed discussion on how SAMOSA learns to effectively filter OODs and explained that informativeness drives OSAL performance more than pure OOD filtering (supported by [1] and our findings). We updated the paper accordingly.
- Asked about the L1 norm between $f_{SAM}(x)$ and $f_{SGD}(x)$. We linked this to Theorem 2.1, noting it reflects SAM’s ability to learn flatter minima for robustness and generalization.
- Requested experiments on “real open-set noise” and “0% mismatch ratios,” which we clarified as ambiguous: our experiments already model realistic open-set conditions with varying mismatch ratios and label noise. A 0% mismatch ratio reduces to classic AL, where informativeness alone matters (precisely what SAMOSA optimizes).

**Reviewer B1Uz - initial score: 6, positive trend**
- Positive on motivation, evaluation, and clarity.
- Raised novelty concerns, which we addressed by detailing SAMOSA’s contributions: (i) formalizing the link between loss sharpness and atypical examples, (ii) introducing a SAM-based sampling metric, (iii) offering insights into multi-round dynamics, and (iv) leveraging disagreement grounded in loss geometry to achieve state-of-the-art OSAL performance and improved robustness.

**Reviewer T42G - initial score: 2, raised it to 4 just before the platform issue and asked for more experiments**
- Concern on Figure 3 entropy-based experiments was due to a misunderstanding. We clarified and updated the paper for clarity. The reviewer acknowledged his/her mistake.
- Pointed out a wording issue in Figures 1b and 1c. We revised the paper accordingly. Nevertheless, the core message remains: uncertainty-based OSALs miss confidently misclassified samples, while purity-based OSALs select uninformative ones. Figure 1 shows that SAMOSA tackles both issues and the reviewer agreed, removing this as a concern.
- Similar to DZBW, the reviewer asked about OOD filtering. We provided a detailed explanation and updated the paper accordingly. The reviewer was satisfied with our explanation.
- Asked about SAMOSA as inconsistency-based sampling. We clarified this aspect in Appendix A.9: SAMOSA is inspired by inconsistency-based approaches but its gains stem from a geometric mechanism that better distinguishes informative samples from non-informative ones, namely SAM-SGD gap (Theorem 3.1). Table 3 confirms it by showing SAMOSA outperforms a purely inconsistency-based SAMOSA alternative.
- Asked about performance gains in Figure 4. We provided detailed numerical analysis in Tables 1, 5, and 6 (SAMOSA’ gain is up to 3%).
- The reviewer raised his score satisfied with our responses to the above concerns and still had concerns about our algorithm being just an inconsistency-sampling and asked for more experiments. We implemented the most recent and strongest baseline (EAOA published in June 2025), and show that SAMOSA outperforms it in almost all settings we tried, reinforcing its state-of-the-art OSAL status. We updated the paper accordingly. We believe that the additional experimental evaluation addresses the reviewer concerns.

The discussion was very constructive, and we believe all major concerns were resolved. The paper has improved with additional experiments and a clearer presentation of SAMOSA’s motivations. Based on reviewer interactions, we believe that the scores would likely have increased if the discussion had not been cut short. We remain available for any further questions and would like to thank you for your time under these unusual circumstances.

---

### Meta-Review · Area_Chair_QUNh · 2026-01-15

**Summary:**

The paper proposes SAMOSA for open-set active learning, using the SAM–SGD prediction gap as a typicality/informativeness score, and reports consistent but modest gains (often ~2–3%) on standard OSAL benchmarks. Reviewers generally agreed the idea is reasonable and the empirical package is solid, but the main concerns were (i) novelt: whether this is essentially an inconsistency/disagreement sampler with SAM as a training trick, (ii) clarity/strength of the theoretical justification (esp. why the L1 gap is the right OSAL signal), and (iii) external validity: evaluation is largely restricted to synthetic open-set splits of CIFAR/TinyImageNet with limited evidence beyond these settings. Overall, the contribution feels incremental relative to prior disagreement-based OSAL work, and the improvements, while consistent, are not decisive enough to overcome the novelty/validation concerns.

**Reviewer Concerns:**

Several concerns were addressed well: the confusion around entropy-based evidence (Fig. 3) and the wording issue around purity-vs-uncertainty (Fig. 1) were clarified and corrected; the authors added stronger baseline coverage by implementing EAOA and showed SAMOSA is competitive or better in most settings; and they provided more discussion/metrics around sampling precision and the purity–informativeness tradeoff over rounds.

Remaining concerns are still meaningful: the core novelty question (is this fundamentally more than a standard inconsistency criterion plus SAM?) was mitigated but not fully resolved; the theoretical story (Theorem 3.1 / mechanism-level necessity of SAM vs generic disagreement) remains somewhat reliant on narrative and limited ablations; and the experimental scope is still mostly standard synthetic OSAL splits without a clearly stronger “realistic” open-set/noise setting or broader-domain validation.

**Reviewer Scores:**

With full discussion, I would not expect large upward movement. DZBW (6) likely stays at 6 (borderline positive but still “would not mind rejection”), as the added precision discussion helps but doesn’t fully resolve the external-validity and motivation gaps. B1Uz (6) explicitly indicated they would retain 6. T42G moved from 2→4 after rebuttal; with the added EAOA baseline and clarifications they likely remain around 4–5, but their main remaining concern about incremental novelty versus inconsistency-based methods would likely persist. Net effect: the score profile remains mixed and not clearly above threshold.

---

### Decision · Program_Chairs · 2026-01-26

Reject